# End-to-End Ontology Learning with Large Language Models

**Andy Lo**
University of Cambridge
cyal4@cam.ac.uk

**Albert Q. Jiang**
University of Cambridge
qj213@cam.ac.uk

**Wenda Li**
University of Edinburgh
wenda.li@ed.ac.uk

**Mateja Jamnik**
University of Cambridge
mateja.jamnik@cl.cam.ac.uk

## Abstract

Ontologies are useful for automatic machine processing of domain knowledge as they represent it in a structured format. Yet, constructing ontologies requires substantial manual effort. To automate part of this process, large language models (LLMs) have been applied to solve various subtasks of ontology learning. However, this partial ontology learning does not capture the interactions between subtasks. We address this gap by introducing OLLM, a general and scalable method for building the taxonomic backbone of an ontology from scratch. Rather than focusing on subtasks, like individual relations between entities, we model entire subcomponents of the target ontology by finetuning an LLM with a custom regulariser that reduces overfitting on high-frequency concepts. We introduce a novel suite of metrics for evaluating the quality of the generated ontology by measuring its semantic and structural similarity to the ground truth. In contrast to standard syntax-based metrics, our metrics use deep learning techniques to define more robust distance measures between graphs. Both our quantitative and qualitative results on Wikipedia show that OLLM outperforms subtask composition methods, producing more semantically accurate ontologies while maintaining structural integrity. We further demonstrate that our model can be effectively adapted to new domains, like arXiv, needing only a small number of training examples. Our source code and datasets are available at https://github.com/andylolu2/ollm.

## 1 Introduction

An ontology is a formal and structural way of representing domain-specific concepts and their relations [16]. They can be simple (e.g., Wikipedia categories) consisting of *concepts* and only a small number of types of *taxonomic relations* (e.g., *is-a* relationships), or they can be complex (e.g., Schema.org) consisting of axioms or many types of relations. For example, a simple ontology for programming languages might contain two concepts "Dynamically-typed language" and "Python", and one relation "Dynamically-typed language $\rightarrow$ Python", representing the knowledge that Python is a dynamically-typed language. A more complex ontology might contain axioms too, for example, "all programming languages are either dynamically or statically typed". In this paper, we focus on ontologies with only concepts and taxonomic relations. Compared to typical deep learning models, which represent knowledge implicitly in its weights, ontologies capture knowledge in a structured and explicit manner, making them reliable, easy to edit and human-interpretable. Such benefits of ontologies have led to their wide adoption in practice. For example, Wikipedia categories have been

38th Conference on Neural Information Processing Systems (NeurIPS 2024).

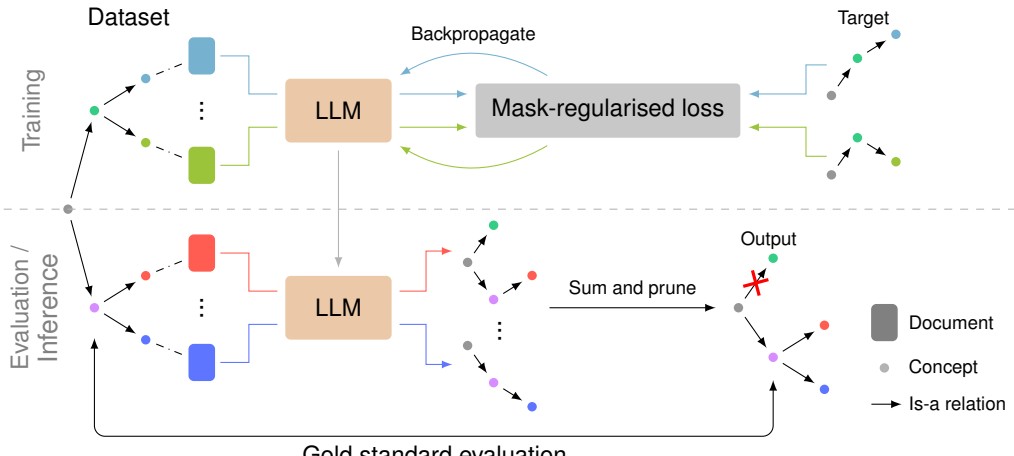

Figure 1: OLLM: Using annotations of documents with their relevant concepts, we train an LLM to model relevant subgraphs of the target ontology with a custom regulariser. During inference, the generated subgraphs for each document are summed and pruned to give the final output ontology. For evaluation, we measure the similarity between the generated ontology and the ground truth.

used for entity ranking [46] and information retrieval [42], or Schema.org [40] is a core component of the Semantic Web [1] initiative.

While ontologies are useful, building ontologies often requires substantial manual effort. Ontology learning (OL) is the study of automating the construction of high-quality ontologies at scale. For a simple ontology, this amounts to discovering the concepts and taxonomic relations, usually based on a source corpus. In this paper we aim to develop domain-independent methods for OL that are scalable and produce better ontologies.

Traditionally, OL is viewed as a composition of subtasks [3], such as concept discovery and relation extraction. In particular, prior works have demonstrated that state-of-the-art large language models (LLMs) can solve such subtasks effectively [4]. While studying subtasks permits fine-grained analysis and evaluation, it does not directly indicate the subsequent impact on the quality of the final ontology. Moreover, there is potential room for improvement by combining several subtasks into one, such as by modelling concepts and relations in conjunction. In this paper, we instead develop and evaluate methods that construct ontologies in an end-to-end fashion to answer the following research questions:

1. How can we leverage LLMs' knowledge base to build ontologies from scratch?
2. Does our method scale efficiently to practical problem sizes?
3. How well does our method generalise to new domains?

We introduce OLLM, an end-to-end method for using LLMs to construct ontologies at scale. Rather than focusing on individual relations between concepts, we finetune an LLM to model entire sub-components of the target ontology. The output ontology is generated by taking the sum of generated sub-components and applying simple post-processing. An overview of the pipeline is shown in Figure 1. To train OLLM, we collect the categorisation metadata for a subset of Wikipedia articles. We attempt to adapt an LLM to model the relevant categorisation subgraph for a particular Wikipedia article, but discover that direct finetuning leads to poor generalisation due to overfitting to high-level, frequently occurring concepts. Instead, we propose a custom regulariser that reweights each concept based on its frequency of occurrence, which substantially improves generalisation.

We evaluate OLLM by measuring the similarity of the generated ontology with the ground truth. Current approaches for comparing ontologies rely on mapping components of the two ontologies onto each other, most commonly by literal text matching [30, 45]. This is unreliable when the two ontologies are not already sufficiently similar. Instead, we propose a suite of evaluation metrics suitable for comparing arbitrary labelled graphs. These metrics compare edges and subgraphs of the two ontologies using pretrained text embedders to test for semantic and structural similarity. Both our quantitative and qualitative results reveal that an LLM can already outperform existing

extraction-based methods out of the box, and the performance is further improved by finetuning with our custom regulariser. We additionally demonstrate that OLLM can be adapted to build the arXiv ontology using only a small number of training examples, suggesting that our model can be applied to new domains in a data-efficient way. In summary, our contributions are:

1. We constructed two datasets based on Wikipedia and arXiv, which can serve as standard datasets for future work studying end-to-end OL.
2. We created OLLM, a method that utilises LLMs to build ontologies from scratch. OLLM produces high-quality ontologies and serves as a strong baseline for end-to-end OL.
3. We developed new evaluation metrics for assessing the quality of the generated ontologies.

## 2 Background

An ontology is a structured way of representing concepts and relations of a shared conceptualisation, that is, domain knowledge [15, 16]. Ontologies can span a wide range of complexities. A fully-fledged ontology might contain concepts, relations, constraints, and axioms that enable complex automated reasoning. In this paper, we focus on the core building blocks of an ontology: concepts and taxonomic relations which represent *is-a* or *is-subclass-of* relationships between concepts. In some cases, the *is-part-of* relation is also considered a taxonomic relation. We treat such an ontology as a rooted labelled directed graph where nodes represent concepts, edges represent taxonomic relations and the root node is the special concept of all concepts. A strict ontology asserts that the taxonomic relation is asymmetric and thus the graph must be acyclic, though in practice some ontologies, such as the Wikipedia ontology studied in this paper, may contain cycles. We therefore do not assume that an ontology graph is necessarily acyclic. Examples of ontologies include WordNet [33] with 117,659 concepts and 89,089 taxonomic relations, and the Gene Ontology [2] with 42,255 concepts and 66,810 taxonomic relations.

Ontology learning is the automatic extraction of ontological elements [17]. The most studied source of input is unstructured text, though there are also works on semi-structured data like HTML [22]. In this paper, the input is a set of documents, each consisting of some unstructured text. We additionally assume each document is associated with one or more concepts in the ground truth ontology, which we utilise for training. The goal is to reconstruct the ground truth ontology given the set of documents.

Prior works view OL as a composition of subtasks, and study each subtask in isolation [3, 6]. A typical pipeline for building a simple ontology is to first perform concept discovery (identify the nodes), and then relation extraction (identify the edges) [8, 24]. A notable approach for relation extraction is Hearst patterns [18]. Hearst patterns are hand-crafted lexico-syntactic patterns that exploit natural language structure to discover taxonomic relations. For example, the pattern "[noun phrase] such as [noun phrase]" matches phrases like "dogs such as chihuahuas", and thus can be processed by regular expressions to identify the relation "dog $\rightarrow$ chihuahua". Hearst patterns suffer from low recall, as the relations must occur in exact configurations to be identified by the rules. Roller et al. [39] suggest smoothing techniques to alleviate this issue though at the cost of lower precision.

Recently, language models have been used for OL. REBEL [7] treats relation discovery as a translation task, and finetunes encoder-decoder LLMs to extract both taxonomic and non-taxonomic relations. Babaei Giglou et al. [4] benchmarked a wide family of LLMs for concept and relation discovery, and showed promising results. However, the quadratic complexity of link prediction makes this approach unscalable to large ontologies. We provide more discussion in Appendix A.2.3. There are also proof-of-concept works for building ontologies end-to-end with LLMs. Funk et al. [13] proposes to build an ontology by recursively prompting LLMs, while Trajanoska et al. [44] generate the entire ontology in one completion. However, both studies are limited in the scale of the task and evaluation: they only considered ontologies of up to 1000 concepts and relied on manual qualitative evaluation. We bridge this gap by proposing a method that can scale to practical problem sizes and new metrics for systematic qualitative evaluation.

The evaluation of ontologies is an open research area. The main approaches are gold standard evaluation [51], which matches elements of the generated ontology with a predefined target ontology; task-based evaluation [36], which measures the usefulness of the ontology on a specific application; and human evaluation [5, 37]. In this paper, we evaluate by the gold standard metric as it is the most straightforward approach when ground-truth ontology exists. Prior works have considered

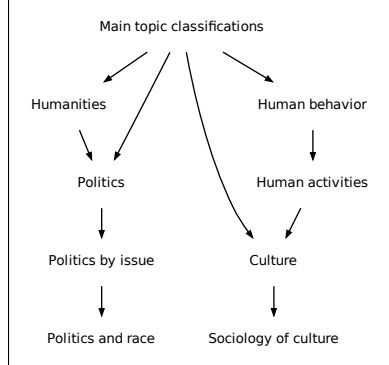

```
[INST] Title: Hybridity
Hybridity, in its most basic sense ... [/INST]
Main topic classifications -> Human behavior
    -> Human activities -> Culture ->
    Sociology of culture
Main topic classifications -> Humanities ->
    Politics -> Politics by issue -> Politics
    and race
Main topic classifications -> Politics ->
    Politics by issue -> Politics and race
Main topic classifications -> Culture ->
    Sociology of culture
```

Figure 2: Example subgraph induced for the Wikipedia page "Hybridity" (left), where $N = 4$ and $C = \{\text{Politics and race}, \text{Sociology of culture}\}$. The corresponding training text sequence (right), where text coloured in grey is ignored as training targets, but is still present as context for later tokens.

matching concepts [30] and direct or indirect relations [23, 45] by literal text comparison. Others have also considered edit-distance [12] or bag-of-words distributional similarity for text comparison [51]. These techniques for measuring semantic similarity may be considered unreliable and have been superseded by current methods [9]. We instead rely on more modern techniques like pretrained text embedders [10] and graph convolutions [26] to match substructures between the two ontologies.

## 3  OLLM

We now introduce OLLM, our novel, simple and scalable method for end-to-end OL with LLMs. On a high level, OLLM uses an LLM to model concept subgraphs of the target ontology by utilising a linearisation scheme to transform subgraphs into string sequences. In contrast to learning individual edges, modelling subgraphs allows the model to learn higher-order structures, such as the interactions between three or more nodes. To create the training dataset, OLLM relies on the annotations of documents to concepts to generate document-subgraph pairings. Such subgraphs are much smaller than the complete graph, so they can be learned by the model more easily. The generated subgraphs for each document are summed into a weighted graph, and simple post-processing is applied to obtain the final predicted ontology.

### 3.1  Subgraph modelling

Here, we describe the method for creating document-subgraph pairings. Given a document and its associated set of concepts $C$, we define the *relevant paths* as the set of paths of at most length $N$ from the root to any of the concepts in $C$. The *relevant subgraph* is the set of nodes (concepts) and edges (taxonomic relations) that occur at least once in the relevant paths. An example is shown in Figure 2 (left). The choice of $N$ is task-specific and we describe our method for choosing $N$ in Section 5.1.

To employ LLMs to model the subgraphs, we must linearise the graph into a string sequence. Existing methods for autoregressive graph generation employ BFS [50] or DFS [14] ordering starting at an arbitrary node. We instead choose to linearise the subgraph as a list of relevant paths that produced the subgraph in the first place. We do so over BFS/DFS ordering for three reasons: 1) the subgraph is defined from the relevant paths, which makes them the most natural representation; 2) we hypothesise that the hierarchy of concepts in each path is a desirable inductive bias for the hierarchical nature of an ontology; and 3) the path-based representation is much easier to describe in natural language instructions so that our LLM prompting-based baselines may produce reasonable results without finetuning. The linearisation template can be found in Figure 5 in Appendix A.1.2.

### 3.2  Post-processing

The final output graph is obtained by summing all generated subgraphs for each document and pruning low-weighted components. Given the generated subgraphs $G_1 = (V_1, E_1), \ldots, G_n = (V_n, E_n)$, the raw output graph is defined as $G_{\text{raw}} = (V_{\text{raw}}, E_{\text{raw}})$, where $V_{\text{raw}} = \cup_{i=1}^n V_n$ and $E_{\text{raw}} = \cup_{i=1}^n E_n$.

Each edge $(u, v) \in E_{\text{raw}}$ is additionally weighted by the number of times it occurs in the collection of subgraphs: $w(u, v) = \sum_{i=1}^{n} \mathbb{1}[(u, v) \in E_n]$. A few simple post-processing steps are then applied to $G_{\text{raw}}$ in order to prune it:

1. Self-loop pruning: All edges $(u, u) \in E_{\text{raw}}$ are removed.
2. Inverse-edge pruning: For $(u, v) \in E_{\text{raw}}$, if $(v, u) \in E_{\text{raw}}$ and $w(v, u) > w(u, v)$, remove $(u, v)$. That is, bidirectional edges are turned into unidirectional ones.
3. Absolute thresholding: Edges in $E_{\text{raw}}$ with weight below the $\alpha$-th quantile are removed, where $0 \leq \alpha \leq 1$ is a hyperparameter. This removes edges that are globally less important.
4. Relative thresholding: For each vertex $u \in V_{\text{raw}}$, let $e_1, \ldots, e_k$ be the outgoing edges from $u$, sorted by weight in ascending order. Let the cumulative weight be $C(e_i) = \sum_{j=1}^{i} w(e_j) / \sum_{j=1}^{k} w(e_j)$. The edges $\{e_i \mid C(e_i) \leq \beta\}$ are pruned, where $0 \leq \beta \leq 1$ is a hyperparameter. This is similar to top-$p$ sampling [19], which we use to remove edges that are less important than their neighbours.
5. Clean up: After pruning all edges, nodes with no incoming or outgoing edges are removed.

We choose the hyperparameters $\alpha$ and $\beta$ by tuning on the validation set (Section 5.1).

## 4 Evaluating end-to-end OL

Ontology evaluation is a hard problem as there are no quantitative definitions of what constitutes a "good ontology", and metrics generally only capture one aspect (e.g., structure but not semantics) of an ontology. We approach evaluation by treating the ground truth as a proxy for a good ontology, and comparing the generated ontologies against the ground truth. Here, we describe how the ground truth is obtained, and introduce new evaluation metrics that are used for measuring ontology similarity.

### 4.1 Dataset

We collect the datasets for the two ontologies considered in this paper: Wikipedia categories and the arXiv taxonomy. We use Wikipedia for learning and in-domain evaluation, and arXiv for out-of-domain evaluation. To build the Wikipedia dataset, we perform a BFS traversal from its root category "Main topic classifications" up to depth 3. For every category encountered, we retrieve the titles and summaries (the text before the first section) of up to 5000 pages that belong in that category. The source data is obtained from the Wikipedia API.[1] The arXiv taxonomy is available from its home page, and the source corpus is constructed from the title and abstract of all the papers uploaded to arXiv in the years 2020–2022 with more than or equal to 10 citations.[2] In total, the Wikipedia dataset has 13886 concepts, 28375 taxonomic relations and 362067 documents, while the arXiv dataset has 161 concepts, 166 taxonomic relations and 126001 documents.

Generating the train and test splits from the datasets is a non-trivial problem. Each training example consists of a document and its relevant subgraph (Section 3.1). The naive approach of randomly selecting a subset of document-subgraph pairs for the training likely leads to data leakage as there might be a significant overlap between subgraphs in the training set and the test set. Instead, we first split the full ontology into train and test graphs, and then generate the training document-subgraph pairs. This ensures that there are sufficiently many unseen concepts (and thus relations) in the test split, as shown in Figure 3. Our method is as follows:

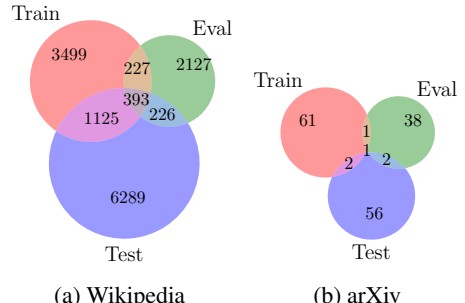

(a) Wikipedia  (b) arXiv

Figure 3: Intersection of concepts among the train, validation and test splits of the datasets.

1. Let $V^{\text{top}}$ be the set of top-level nodes, that is, children of the root node. Randomly partition $V^{\text{top}}$ into train $V_{\text{train}}^{\text{top}}$, validation $V_{\text{val}}^{\text{top}}$, and test $V_{\text{test}}^{\text{top}}$ splits in 7:3:10 ratio.

---

[1] https://en.wikipedia.org/w/api.php
[2] Citation counts obtained from https://api.semanticscholar.org/.

2. Let $d$ be the depth on the full graph, that is, the distance of the furthest node from the root. The nodes of the train graph are taken as the union of all the nodes that are within distance $d - 1$ from any node in $V_{\text{train}}^{\text{top}}$, plus $V_{\text{train}}^{\text{top}}$ and the root. The edges are all the edges in the full graph that have both endpoints in the train graph. Similar applies for $V_{\text{val}}^{\text{top}}$ and $V_{\text{test}}^{\text{top}}$.

## 4.2 Metrics

Many existing methods for comparing ontologies rely on syntactic measures like string edit distance [12] as a proxy for semantic similarity, or require every concept to be tagged with descriptions or documents for distributional semantics comparison [51]. To obtain more robust and general evaluation results, we introduce a suite of similarity metrics that use modern methods like text embeddings [38]. Multiple metrics are used as they trade off between interpretability and comprehensiveness and we aim to make them complementary by capturing different aspects of an ontology. For example, comparing ontologies by literal text equality is easy to understand but may be unreliable. In Section 5.4, we provide further discussion on evaluation metrics in the context of our experiment results. We denote the ground truth ontology graph as $G = (V, E)$ and the generated graph as $G' = (V', E')$.

**Literal F1** While literal text matching is unreliable, it is also the simplest and the most interpretable. We treat this metric as a reference metric for sanity check. The Literal F1 metric [23] is given by the harmonic mean of the precision and recall of the edges:

$$\text{Literal precision} = \frac{|E \cap E'|}{|E'|} \qquad \text{Literal recall} = \frac{|E \cap E'|}{|E|}$$

**Fuzzy F1** The Literal F1 metric puts a strong emphasis on using the correct wording, while in practice, we are interested in evaluating the semantics of an ontology. For example, using a synonymous phrase for a concept should not be penalised. We utilise embeddings from a pretrained sentence transformer [38] and use the cosine similarity of the embeddings to measure semantic similarity. Specifically, let $\text{NodeSim}(u, u') \in V \times V' \to [-1, 1]$ be the cosine similarity between the sentence embeddings for $u$ and $u'$. The Fuzzy F1 score is obtained from the fuzzy precision and recall, defined as:

$$\text{Fuzzy precision} = \frac{|\{(u', v') \in E' \mid \exists (u, v) \in E. \text{NodeSim}(u, u') > t \land \text{NodeSim}(v, v') > t\}|}{|E'|}$$
$$\text{Fuzzy recall} = \frac{|\{(u, v) \in E \mid \exists (u', v') \in E'. \text{NodeSim}(u, u') > t \land \text{NodeSim}(v, v') > t\}|}{|E|}$$

where $t$ is the matching threshold. We use all-MiniLM-L6-v2 [38, 47] as the embedding model, and choose $t$ as the median cosine similarity between the synonyms in WordNet [33], computed as 0.436.

**Continuous F1** With fuzzy comparisons, the matches between the edges of the generated and the ground truth graph are no longer one-to-one. This is problematic: consider two graphs $A \to B$ and $B \leftarrow A \to B'$, where $B$ and $B'$ match fuzzily. Such graphs will achieve a perfect Fuzzy F1 score yet they significantly differ. Additionally, we found that the previous metrics fail to provide a useful signal for hyperparameter tuning, particularly for our baselines where the generated graphs are poor. The Continuous F1 metric solves these issues by computing the highest-scoring edge matching between the two graphs, where the similarity score between $(u, v)$ and $(u', v')$ is given by $\min(\text{NodeSim}(u, u'), \text{NodeSim}(v, v'))$. Obtaining such matching is equivalent to solving the linear assignment problem [32], which can be computed by the Hungarian algorithm [27]. The Continuous F1 score is obtained from the continuous precision and recall, given by:

$$\text{Continuous precision} = \frac{s_{\text{cont}}}{|E'|} \qquad \text{Continuous recall} = \frac{s_{\text{cont}}}{|E|}$$

where $s_{\text{cont}}$ is the score achieved by the best edge matching.

```
[INST] Title: List of general awards in the humanities
This list of general awards in the humanities ... from that country. [/INST]
Main topic classifications -> Society -> Humanities -> Humanities awards
Main topic classifications -> Academic disciplines -> Humanities -> Humanities awards
Main topic classifications -> Society -> Culture -> Cultural lists -> Lists of awards
Main topic classifications -> Culture -> Cultural lists -> Lists of awards
Main topic classifications -> Lists -> Cultural lists -> Lists of awards
Main topic classifications -> Humanities -> Humanities awards
Main topic classifications -> Academic disciplines -> Liberal arts education -> Humanities -> Humanities awards
```

(a) Direct finetuning

```
[INST] Title: List of general awards in the humanities
This list of general awards in the humanities ... from that country. [/INST]
Main topic classifications -> Society -> Humanities -> Humanities awards
Main topic classifications -> Academic disciplines -> Humanities -> Humanities awards
Main topic classifications -> Society -> Culture -> Cultural lists -> Lists of awards
Main topic classifications -> Culture -> Cultural lists -> Lists of awards
Main topic classifications -> Lists -> Cultural lists -> Lists of awards
Main topic classifications -> Humanities -> Humanities awards
Main topic classifications -> Academic disciplines -> Liberal arts education -> Humanities -> Humanities awards
```

(b) Finetuning with masked loss

Figure 4: Per token loss on a test set example of the final model trained with and without the custom masked loss objective. A stronger red colour represents a higher cross-entropy loss. Within the top-level concepts (children of the root) shown here, "Culture" and "Humanities" are in the training set while others are not. Using the masked loss objective improves generalisation on the high-level relations (e.g., "Main topic classifications" $\rightarrow$ "Academic disciplines") while maintaining performance on lower-level relations.

**Graph F1** Instead of individual edges, this metric aims to capture the wider structure of the two graphs. Intuitively, we want to know how concepts are related to their local neighbourhood. We do so by using simple graph convolutions [49] with $K = 2$ to compute graph-aware node embeddings after embedding each node with the pretrained embedder. Such embeddings in $G$ are compared against those in $G'$ by cosine similarity, and the highest-scoring node matching, similar to the Continuous F1 metric, gives the graph similarity score. The Graph F1 score is computed from the graph precision and recall, defined as:

$$\text{Graph precision} = \frac{s_{\text{graph}}}{|V'|} \qquad \text{Graph recall} = \frac{s_{\text{graph}}}{|V|}$$

where $s_{\text{graph}}$ is the score achieved by the best node matching.

**Motif distance** Taking inspiration from classical network analysis, we use *network motifs* [34, 41] to evaluate the structural integrity of the generated graphs. Network motifs are reoccurring subgraphs in a larger graph, most commonly 3-vertex subgraphs. They are typically indicative of the structural characteristics of the full graph. We define the motif distance as the total variation distance between the distribution of all 3-vertex subgraphs in $G$ and $G'$.

## 5 Experiments

We design our experiments to answer the following research questions:

1. Does OLLM produce better ontologies than traditional methods by subtask composition?

2. Can OLLM be easily adapted to a new domain?

We approach the questions by training OLLM on the Wikipedia dataset, and further transfer the model to arXiv with a small number of arXiv samples. As baselines, we use two relation extraction methods, Hearst patterns [18, 39] and REBEL [7]. Relation extraction depends on successful concept discovery to produce high-quality ontologies. To estimate a ceiling to such baselines, *we give the baselines a substantial advantage* by providing them with the ground truth concepts in the test graph. The results show that even with such an advantage, OLLM outperforms the baselines on many metrics, demonstrating the potential of OLLM for end-to-end OL (Section 5.3).

## 5.1   Implementation details

Analysing the per-token loss on the test split sequences of a directly finetuned model (Section 3.1) shows that the model tends to memorise high-level relations from the training set, leading to poor generalisation, as shown in Figure 4 (top). The crux of the problem is that low-level relations are substantially more diverse than high-level ones: since we present both types of relations at the same rate to the model, it tends to overfit on high-level relations while underfitting on low-level ones. To alleviate this issue, we introduce a new training objective that randomly masks the loss contribution of frequently occurring relations. Suppose a relation $u \rightarrow v$ is present $n$ times in the training set. During training, when $u \rightarrow v$ appears in one of the relevant paths, we mask the loss contribution of the tokens for $v$ with probability $\max(1 - M/n, 0)$, where $M$ is a constant for the average number of times a relation is present in the training set. Intuitively, this regulariser ensures that frequent relations are only seen $\approx M$ times as targets throughout training, hence reducing overfitting as shown in Figure 4 (bottom). Note that while $v$ is masked from the target, its tokens are still present in the input sequence as context for later tokens. A concrete training example can be found in Figure 2 (right).

We finetune Mistral 7B v0.2 [21] with Low-Rank Adaptation [20] on the masked loss objective. The model is trained on the Wikipedia dataset for two epochs with Adam [25]. During inference, the outputs are generated with temperature 0.1 and nucleus sampling [19] top-$p$ of 0.9. We include a finetuning baseline without the masked loss objective, denoted as Finetune. To adapt OLLM for arXiv, we further finetune the model on 2048 document-subgraph pairs from arXiv. We initialise new low-rank adaptors and train until the loss stops improving on the validation set. We name these models OLLM (transfer) and Finetune (transfer) for training with and without the masked loss objective, respectively. Full details for the Wikipedia and arXiv experiments can be found in Appendix A.1.2.

The hyperparameters for the post-processing steps are tuned by grid search on the validation set. We sweep over $\alpha \in 1 - \text{geomspace}(1/|E_{\text{raw}}|, 1, 21)$ and $\beta \in \text{geomspace}(0.1, 1, 21) - 0.1$, and use the values that maximise Continuous F1. For Wikipedia, we choose the subgraph modelling path length $N = 4$ as it is the smallest $N$ such that almost all edges ($> 99\%$) occur in at least one relevant subgraph. Such criterion is used since smaller $N$ results in smaller subgraphs, which we expect to be easier to model accurately. We choose $N = 3$ for arXiv for the same reason.

## 5.2   Baselines

We give a brief overview of the baseline methods here (in addition to Finetune and Finetune (transfer)). The full implementation details can be found in Appendix A.1. All baselines produce weighted directed graphs which we apply the same post-processing steps as in OLLM (Section 3.2) to obtain the final predicted graph.

**Memorisation**  Simply memorising the train graph is a surprisingly strong baseline due to the overlap between train and test graphs, especially for Wikipedia. The weight of each edge is given by the number of relevant subgraphs in which it appears.

**Hearst**  We follow the improved implementation of Hearst patterns by Roller et al. [39]. The authors propose spmi, a method which uses low-rank approximations to smooth the relation matrix so that two concepts can be compared even if there are no direct matches between them. We use the smoothed relation matrix to weigh the relations between the ground truth concepts. The additional hyperparameter for the rank of the smoothed matrix is tuned by grid search over the validation set.

**REBEL**  The REBEL-large model [7] is an LLM trained to extract many types of relations from Wikipedia articles. We only take the "subclass of", "instance of", "member of" and "part of" relations that were extracted. Similar to Hearst, we find that it fails to find many direct relations between ground truth concepts. The same low-rank smoothing technique is applied to improve recall.

**Prompting**  We test the Zero/One/Three-shot performance of instruction-tuned LLMs on the subgraph modelling task described in Section 3.1. To obtain more comparable results, we use Mistral 7B Instruct v0.2, the instruction-tuned version of the base model of OLLM, as the LLM for our prompting baseline. The prompt template used is shown in Figure 6 in Appendix A.1.

**Finetune**  To test the effectiveness of our masked-loss objective, we introduce a direct finetuning baseline using the same configuration as OLLM except it is trained without loss masking.

Table 1: Evaluation metrics of OLLM and baselines on Wikipedia and arXiv. OLLM performs particularly well in modelling semantics, and remains competitive syntactically and structurally.

| Dataset | Method | Literal F1 ↑ | Fuzzy F1 ↑ | Cont. F1 ↑ | Graph F1 ↑ | Motif Dist. ↓ |
|---|---|---|---|---|---|---|
| Wikipedia | Memorisation | **0.134** | 0.837 | 0.314 | 0.419 | 0.063 |
| | Hearst | 0.003 | 0.538 | 0.350 | 0.544 | 0.163 |
| | Rebel | 0.004 | 0.624 | 0.356 | 0.072 | 0.132 |
| | Zero-shot | 0.007 | 0.871 | 0.455 | 0.639 | 0.341 |
| | One-shot | 0.031 | 0.888 | 0.477 | 0.610 | 0.314 |
| | Three-shot | 0.031 | 0.880 | 0.475 | 0.622 | 0.354 |
| | Finetune | 0.124 | 0.884 | 0.470 | 0.588 | **0.050** |
| | **OLLM** | 0.093 | **0.915** | **0.500** | **0.644** | 0.080 |
| arXiv | Memorisation | 0.000 | 0.207 | 0.257 | 0.525 | **0.037** |
| | Hearst | 0.000 | 0.000 | 0.151 | 0.553 | 0.098 |
| | Rebel | 0.000 | 0.060 | 0.281 | 0.546 | 0.088 |
| | Zero-shot | 0.025 | 0.450 | 0.237 | 0.414 | 0.145 |
| | One-shot | **0.072** | 0.460 | 0.290 | 0.433 | 0.293 |
| | Three-shot | 0.051 | 0.405 | 0.212 | 0.385 | 0.124 |
| | Finetune (transfer) | 0.000 | 0.440 | 0.225 | 0.441 | 0.148 |
| | **OLLM** (transfer) | 0.040 | **0.570** | **0.357** | **0.633** | 0.097 |

## 5.3 Results

We first evaluate whether OLLM can accurately create ontologies with many concepts and relations, such as the Wikipedia categories. Computationally, OLLM required 12 A100-hours for training and 7 A100-hours for inference to generate an ontology for Wikipedia. This is a modest cost in current standards, which demonstrates the scalability of OLLM for real-world problems. In terms of performance, OLLM produces the most semantically accurate ontology in comparison to our baselines as presented in Table 1. Across all of Fuzzy F1, Continuous F1 and Graph F1, we observe the trend that OLLM scores the best, followed by Finetune and Prompting, and lastly Hearst and REBEL. This is surprising, as it suggests that the combination of LLMs with our subgraph modelling framework is a sufficiently strong inductive bias for LLMs to outperform traditional methods even without finetuning. However, prompting alone is not sufficient to build high-quality ontologies. On the Motif Distance metric, prompting methods score poorly at 0.314–0.354 in comparison to 0.050 and 0.080 for Finetune and OLLM respectively. This shows that using LLMs out-of-the-box for subgraph modelling results in poor structural integrity, though this issue is solved by finetuning. Qualitatively, we observe that OLLM can adhere to the clear, explicit naming style of Wikipedia, even on unseen topics in the test set. For example, it generates "Mathematical categories" and "Groups (mathematics)" under the parent concept "Mathematical structures" to distinguish from the natural language sense of categories and groups (Figure 11c). Such style is not learned by the prompting baselines: Three-shot generated "Elections → France", while it most likely meant "Elections → Elections in France" (Figure 18c). More sample outputs are shown in Appendix A.4.1.

The arXiv task differs from the Wikipedia task as it has much fewer relations, and there is even less overlap between the train and test split. This imposes a great challenge on Finetune and OLLM as they need to generalise with a limited diversity of training samples. Despite such constraints, OLLM is substantially better than other methods in modelling the semantics of the test graph. On the Fuzzy F1, Continuous F1, and Graph F1 metrics, OLLM performs the best among all methods with 0.570, 0.357, and 0.633, significantly higher than the next-best of 0.460, 0.290 and 0.546 respectively. Inspecting the generated ontologies (Appendix A.4.2), we observe that prompting baselines tend to produce repetitive concepts such as "Machine Learning and Artificial Intelligence" and "Artificial Intelligence and Machine Learning" (Figure 27), while Hearst and REBEL put almost all concepts under the same parent concept(s) (Figures 23 and 24). We also found that OLLM's output for arXiv contains concepts from Wikipedia, but restructured in a way that fits the arXiv ontology. For example, "Life sciences" and "Biological evolution" appear in the Wikipedia training set under the same parent category "Life" with no direct links between them. On the generated graph for arXiv, "Life sciences" is instead promoted to one of the top-level concepts with "Biological Evolution" as one of its children, which better fits the "fields of science" style of the arXiv ontology (Figure 20). This demonstrates that OLLM can adapt to produce a new type of ontology by restructuring its learned concepts, all using just a small number of training samples.

In summary, OLLM scores the best or is competitive across all metrics in both tasks, with the notable exception of the Literal F1 metric. We attribute this to the fact that Literal F1 is sensitive to factors like casing and choice of words, and generally only measures syntactic similarity. For example, we see that a suboptimal baseline like Memorisation scores the best on this metric with 0.134 on the Wikipedia task. This reflects that syntactic similarity generally does not entail semantic similarity, so syntax-based metrics should not be used as stand-alone measures for ontology quality.

## 5.4 Meta-evaluation

In this section, we analyse the usefulness of our new metrics for measuring graph similarity and discuss the limitations of existing metrics. On the Wikipedia task, Memorisation, despite being clearly the worst in Continuous F1 and Graph F1, performs the best on Literal F1 and the second-best on Motif Distance. This can be attributed to the fact that Literal F1 is sensitive to semantically insignificant syntactic differences such as casing and word form, and thus when the training and test set has non-trivial overlap (Figure 3), it is biased towards methods that overfit. Similarly, as per the method described in Section 4.1, the data splits are constructed with structural symmetry, hence we expect the train and test splits to have a similar graph structure even though the represented concepts are different. As a result, methods that tend to overfit, for example, Memorisation and Finetune, achieve the best scores on Motif Distance. This demonstrates that Literal F1 and Motif Distance only capture syntactic and structural similarity respectively, and thus should not be used as stand-alone metrics for evaluation.

Analysing the edge and node matchings found by our Continuous F1 and Graph F1 metrics on arXiv reveals that they successfully capture some human intuition on semantic similarity between the two ontologies. In Figures 9 and 10, we visualise the ontology generated by OLLM and the ground truth and observe that semantically similar components in the two graphs indeed get matched. For example, the "Physics" and "Mathematics" clusters in the generated graph get matched with the "Mathematics" cluster in the ground truth, "Data Analysis" and "Information" get matched with "Statistics", "Economics" with "Quantitative Finance", and "Life Sciences" with "Quantitative Biology". This suggests that our edge/node matching procedure is capturing a "semantic graph isomorphism" that allows one to compare similar components in the two graphs, even if they do not exactly share the same concepts. We believe this example of a semantic mapping from one ontology to another is strong evidence that our metrics are capturing meaningful qualities of the ontologies.

## 6 Discussion

**Limitations** We only study and evaluate the construction of simple ontologies with only concepts and taxonomic relations. A potential approach to extend OLLM to produce non-taxonomic relations is to add tags indicating the relation type to each edge when linearising the subgraphs for sequence modelling. New evaluation metrics might also be required to handle multiple types of relations. Another limitation is that the taxonomic relations in the generated ontologies are not necessarily transitive due to the existence of cycles. This is a general problem for many OL methods and there are existing works on cycle removal algorithms for cleaning hierarchies [43, 52]. We ablate this in Appendix A.2.1 and found that the generated ontology can be made consistent by removing a small number of edges. Furthermore, we were unable to fully control for data contamination as the pretraining dataset of Mistral 7B is not publically known. We do, however, observe that the generated ontologies are sufficiently different from the ground truth, indicating that OLLM is not directly remembering samples from its pretraining stage.

**Conclusion** In this paper, we introduce a general method for building ontologies in an end-to-end fashion. We propose a set of metrics for end-to-end OL that measures the semantic and structural similarity between arbitrary labelled graphs. Our model, OLLM, outperforms traditional subtask composition methods in reconstructing the Wikipedia categories, and can be transferred to build ontologies for arXiv after finetuning on a small number of examples. Using LLMs as the backbone for subgraph modelling opens up exciting avenues for future research. For example, one may generate ontologies from corpora with images using vision language models [11].

# 7 Acknowledgements

We thank Dr Thomas Sauerwald for suggesting network motifs as a basis for evaluation. AQJ acknowledges the support of a Peterhouse Graduate Studentship.

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

# A  Appendix / supplemental material

## A.1  Experiment details

### A.1.1  Wikipedia

Some prior works that use Wikipedia categories as a dataset perform additional filtering of concepts considered as "meta-categories" mainly used for page management [35]. We instead decided not to further filter the source data to minimise external bias. We note that it is often not clear-cut whether a Wikipedia category is just for page management. For example, the Wikipedia categories of the form "Lists of [subject]" refer to the special type of articles where the main body is a bullet point/table listing of the subject, which is a useful concept in the Wikipedia domain.

### A.1.2  OLLM

For the Wikipedia experiment, we use Mistral 7B v0.2 (not instruction-tuned) [21] as the base model. We attach LoRA [20] adaptors to all attention and feed-forward layers with parameters $r = 32$ and $\alpha = 16$. The model is trained for 2 epochs ($\approx$ 17K steps) with batch size 16, context length 2048, and is optimised with Adam using a constant learning rate of 1e-5 with warm-up from zero for the first 100 steps. Finetune uses the same configuration. Training takes 12 A100-hours.

For the arXiv experiment, we further finetune the model trained on Wikipedia with masked loss objective on 2048 document-subgraph pairs from the arXiv training set. We merge the LoRA adaptors from the Wikipedia experiment and initialise new ones with $r = 8$ and $\alpha = 8$. The model is trained with batch size 16 and Adam with constant learning rate 3e-6 and warp-up from zero for the first 10 steps. Training terminates when the loss stops improving on the evaluation set, which happened at step 288. Finetune (transfer) uses the same configuration. Early stopping happened at step 192.

For both experiments, we finetune the model with the instruction template similar to that of Mistral 7B instruct v0.2. The format is shown below:

```
[INST]\
Title: {{ title }}
{{ abstract }}[/INST]\
{% for path in paths %}
{{ path | join(" -> ") }}
{% endfor %}\

```

Figure 5: Linearisation template for OLLM training.

For inference, we use the vLLM [28] server which achieves a throughput of $\approx$ 10 documents per second. Inference on the validation and test splits of both datasets takes 12 A100-hours in total.

### A.1.3  Hearst

The Hearst baseline follows the implementation by Roller et al. [39]. Using the tokenization, part-of-speech tagging, lemmatisation, and token regex functionality of the CoreNLP pipeline [31], taxonomic relations are extracted according to the 28 Hearst patterns used by the authors. Processing all documents takes 10 CPU-hours.

Following the spmi method, low-rank smoothing is applied to the relation matrix to allow comparison between any two concepts even if they are not directly related by an extracted relation. The rank of the smoothed matrix, $r$, is a hyperparameter which we tune by sweeping over $r \in \{5, 10, 15, 20, 25, 50, 100, 150, 200, 250\}$ on the validation set. This defines a dense weighted graph as the raw output. Unfortunately, computing Continuous F1 on a dense graph is very slow, especially for Wikipedia. This is because the Hungarian algorithm used for solving the optimal matching between edges has time complexity $O(N^3)$, where $N$ is the number of edges. To bypass this issue, we perform a pre-filtering step of only exporting the top $10|V|$ weighted edges in the smoothed relation matrix, where $|V|$ is the number of nodes in the graph. For the datasets considered,

this density of edges is still much higher than that of the ground truth, and thus, we expect this to have minimal impact on the final output after post-processing.

### A.1.4 REBEL

We use REBEL-large [7] in the implementation. The model is an encoder-decoder transformer based on BART-large [29] with 406M parameters. We sample the model with the default configuration used by Cabot and Navigli [7]. The model is trained to predict 220 types of relations, most of which are not taxonomic relations. We filter the extracted relations and only keep those tagged with "subclass of", "instance of", "member of", and "part of" relation types. The same low-rank smoothing method as Hearst is applied to the raw extractions. Processing all documents takes 3 A100-hours.

### A.1.5 Prompting

To obtain more comparable results, we use Mistral 7B Instruct v0.2, the instruction-tuned version of the base model of OLLM, as the LLM for our prompting baseline. For One-shot and Three-shot, we randomly sample examples from the training set for each query. The output is parsed using regex and results that do not match the regex are discarded. We perform manual prompt engineering by inspecting individual responses. The final prompt template is shown in Figure 6. The total inference cost for all prompting baselines is $\approx 50$ A100-hours.

```
The following is an article's title and abstract. Your task is to assign this
    article to suitable category hierarchy. A category is typically represented by
    a word or a short phrase, representing broader topics/concepts that the article
     is about. A category hierarchy represented by a collection of paths from the
    generic root category "Main topic classifications" to a specific category
    suitable for the article. The topics titles should become more and more
    specific as you move from the root to the leaf.

{% if examples|length > 0 %}
{% for example in examples %}
### EXAMPLE {{ loop.index }} ###
### ARTICLE ###
Title: {{ example['title'] }}
{{ example['abstract'] }}
### END ARTICLE ###
{% for path in example['paths'] %}
{{ path | join(" -> ") }}
{% endfor %}
### END EXAMPLE {{ loop.index }} ###
{% endfor %}
{% else %}
You must answer in the format of:
Main topic classifications -> Broad topic 1 -> Subtopic 1 -> ... -> Most specific
    topic 1
Main topic classifications -> Borad topic 2 -> Subtopic 2 -> ... -> Most specific
    topic 2
...
{% endif %}

### ARTICLE ###
Title: {{ title }}
{{ abstract }}
### END ARTICLE ###

Provide a category hierarchy for the above article. \
{% if examples|length > 0 %}
Use the same format as the examples above.
{% else %}
Use the format described above.
{% endif %}
```

Figure 6: Prompt template used for the Zero/One/Three-shot baselines.

### A.1.6  Hyperparameters

The raw generated outputs of all methods are post-processed with the same scheme as described in Section 3.2. The best hyperparameters for the post-processing step found by grid search on the validation are reported in Table 2.

Table 2: Values of the best hyperparameters found by grid search. $r$ is the rank of the low-rank smoothing, only applicable to Hearst and REBEL. $\alpha = \beta = 0$ means no edges are pruned from the raw output apart from self-loop and inverse edge removal.

| Dataset | Method | $\alpha$ | $\beta$ | $r$ |
|---------|--------|----------|---------|-----|
| Wikipedia | Memorisation | 0 | 0.058489 | - |
| | Hearst | 0.786685 | 0 | 5 |
| | REBEL | 0.872544 | 0 | 20 |
| | Zero-shot | 0.976781 | 0.298107 | - |
| | One-shot | 0.990906 | 0.346684 | - |
| | Three-shot | 0.991955 | 0.530957 | - |
| | Finetune | 0.883848 | 0.058489 | - |
| | OLLM | 0.974330 | 0.025893 | - |
| arXiv | Memorisation | 0.340246 | 0 | - |
| | Hearst | 0.595878 | 0 | 150 |
| | REBEL | 0.836685 | 0 | 100 |
| | Zero-shot | 0.999896 | 0.346684 | - |
| | One-shot | 0.999611 | 0.401187 | - |
| | Three-shot | 0.999851 | 0.298107 | - |
| | Finetune (transfer) | 0.988129 | 0.346684 | - |
| | OLLM (transfer) | 0.983681 | 0.123872 | - |

## A.2 Ablations

In this section, we present the results of our ablations regarding output consistency, the benefits of more advanced prompting techniques, and a comparison against LLMs4OL [4].

### A.2.1 Consistency

A common assumption of taxonomic relations is its transitivity and anti-symmetry. One limitation of many OL methods, including OLLM, is that they do not guarantee that the generated ontology is cycle-free, leading to inconsistent taxonomic relations. To achieve consistency, generic post-processing techniques [43] can be applied to remove such cycles.

We analysed the ontologies generated by OLLM and found only 97 simple cycles in Wikipedia and none in arXiv. Using the greedy algorithm of repeatedly removing the edge that breaks the most simple cycles (a heuristic to the smallest set of edges whose removal makes the graph acyclic), we prune all such cycles and make the ontology consistent by removing just 26 of 10414 edges in Wikipedia. This is surprising considering we did not explicitly optimise our model to satisfy consistency.

### A.2.2 Chain of thought prompting

More sophisticated prompting techniques, such as chain-of-thought (CoT) [48] have been shown to bring significant improvements in LLM inference. We explore whether we can establish strong baselines here by employing CoT in our prompting methods.

We extend the zero-shot prompting method such that prediction now involves two rounds of inference: In the first round, we ask the model to describe the possible relevant concepts for the input document and to explain its reasoning. Then, we ask the model to predict the subgraph in the specified format given the additional, self-generated context. The prompts used are shown below:

We tested the CoT method on Wikipedia and found no significant difference from basic zero-shot prompting, as shown in Table 3. We attribute this to the fact that CoT prompting primarily aims to improve logic and reasoning. We hypothesise that the performance in OL is more dependent on the model's understanding of natural language than its ability to perform multi-step reasoning, hence we do not observe any significant improvement from CoT.

```
The following is an article's title and abstract. Briefly break down the topics (
    both specific and general concepts) relevant to this article. Explain your
    reasoning step by step.

### ARTICLE ###
Title: {{ title }}
{{ abstract }}
### END ARTICLE ###
```

Figure 7: Chain-of-thought first prompt

```
Your task now is to assign this article to a suitable category hierarchy. A category
    is typically represented by a word or a short phrase, representing broader
    topics/concepts that the article is about. A category hierarchy is represented
    by a collection of paths from the generic root category "Main topic
    classifications" to a specific category suitable for the article. The topic
    titles should become more and more specific as you move from the root to the
    leaf.

You must answer in the format of:
Main topic classifications -> Broad topic 1 -> Subtopic 1 -> ... -> Most specific
    topic 1
Main topic classifications -> Broad topic 2 -> Subtopic 2 -> ... -> Most specific
    topic 2
...
```

Figure 8: Chain-of-thought second prompt

### A.2.3   Comparison against LLMs4OL

In this ablation, we evaluate whether the improvement by OLLM is due to the improved methodology (end-to-end modelling) or simply due to the use of LLMs. One way to construct ontologies with LLMs proposed by LLMs4OL is to first prompt LLMs for possible concepts in a document, then link prediction by prompting for a yes/no response. Unfortunately, constructing a baseline from such two subtasks is non-trivial. We encountered significant scalability issues in the link prediction stage as it required $O(n^2)$ inferences. We make two modifications to overcome such limitation:

1. After the concept discovery stage, we only discard all but the $n$ most frequent concepts to limit the number of inferences required during link prediction, where $n$ is the number of concepts in the ground truth.
2. Instead of using zero-shot Mistral 7B as the link predictor, we use a finetuned BERT as the link predictor as it runs much faster. Given that LLMs4OL demonstrated that finetuned models perform much better than zero-shot inference on link prediction, we expect the finetuned BERT to be at least as good, if not better, than zero-shot Mistral 7B on this subtask.

We design this ablation such that it is comparable to zero-shot end-to-end modelling: both use zero-shot Mistral 7B as the backbone, just utilised in different ways. We tested this method on Wikipedia and found that it is worse than zero-shot end-to-end modelling on all metrics except Motif Distance, as shown in Table 4. This is evidence that our end-to-end modelling approach is a clear improvement over traditional subtask-based OL. Not only does LLMs4OL suffer from significant scalability bottlenecks thus unlikely to be scalable to solve large problems, its performance is also worse. The results suggest that we can more effectively and efficiently leverage the capabilities of LLMs beyond just solving subtasks, such as by predicting subgraphs.

Table 3: Comparison of Zero-shot with and without chain-of-thought prompting. There is no significant difference in performance.

| Dataset | Method | Literal F1 ↑ | Fuzzy F1 ↑ | Cont. F1 ↑ | Graph F1 ↑ | Motif Dist. ↓ |
|---------|--------|--------------|------------|------------|------------|---------------|
| Wikipedia | Zero-shot | **0.007** | 0.871 | **0.455** | **0.639** | **0.341** |
| | Zero-shot CoT | **0.007** | **0.873** | 0.449 | 0.635 | 0.357 |

Table 4: Comparison of Zero-shot end-to-end modelling and LLMs4OL-style modelling with zero-shot concept discovery and fine-tuned BERT link prediction. LLMs4OL generally performs worse than zero-shot.

| Dataset | Method | Literal F1 ↑ | Fuzzy F1 ↑ | Cont. F1 ↑ | Graph F1 ↑ | Motif Dist. ↓ |
|---------|--------|--------------|------------|------------|------------|---------------|
| Wikipedia | Zero-shot | **0.007** | **0.871** | **0.455** | **0.639** | 0.341 |
| | LLMs4OL | 0.003 | 0.841 | 0.428 | 0.482 | **0.092** |

## A.3 Visualising evaluation metrics

### A.3.1 Visualisation of node matching in Graph F1

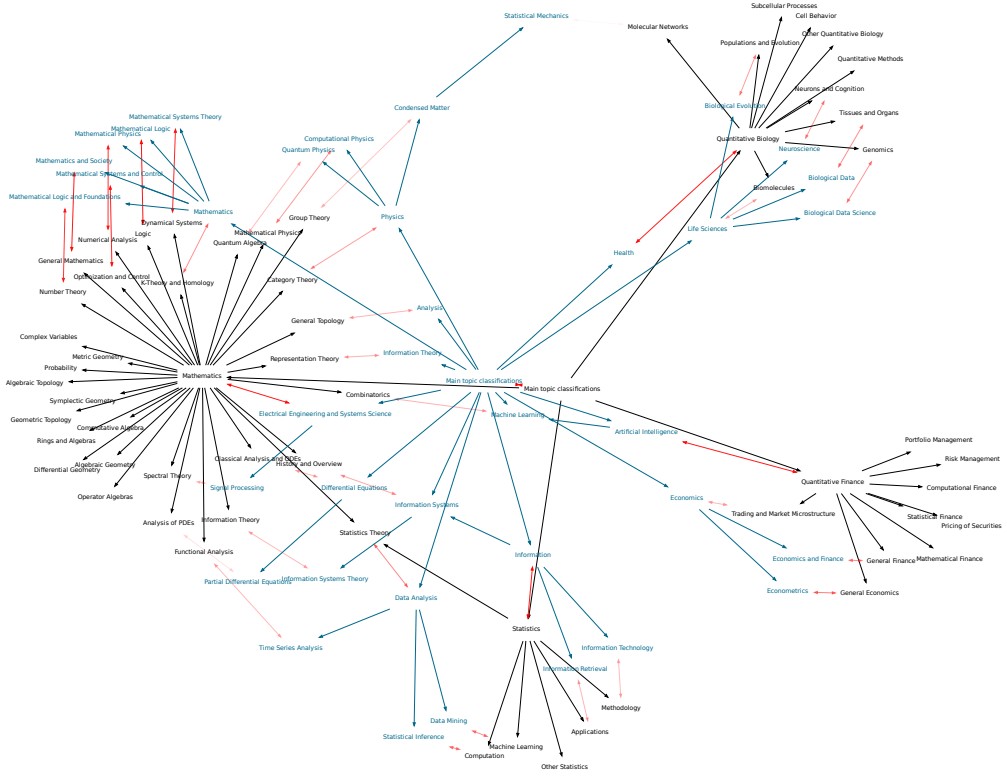

Figure 9: Highest scoring node matching from the Graph F1 metric between the ontology generated by OLLM (**teal**) and the ground truth ontology (**black**). The matching between nodes is shown in **red**, where the opacity of the edge indicates the similarity score (weaker links are more transparent). Visually, the matching defines a clear alignment of the two graphs: from the centre to the left we have the Mathematics-related concepts; at the top right we have Biology-related concepts; and at the bottom right we have Economics-related concepts.

### A.3.2 Visualisation of edge matching in Continuous F1

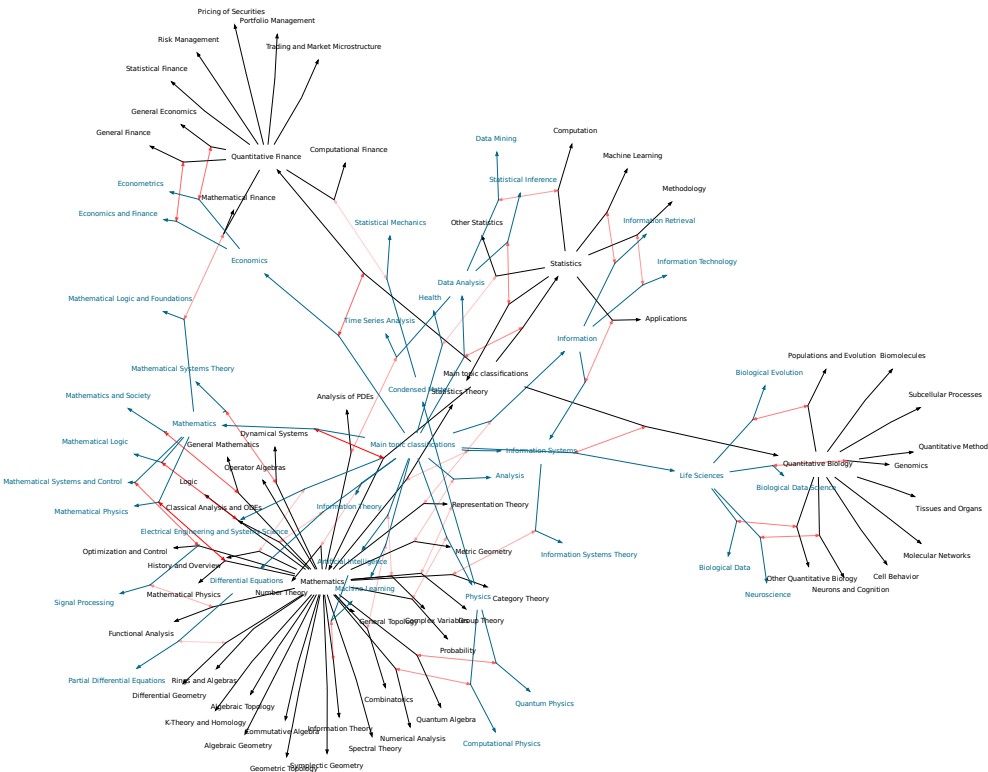

Figure 10: Highest scoring edge matching from the Continuous F1 metric between the ontology generated by OLLM (**teal**) and the ground truth ontology (**black**). The matching between edges is shown in **red**, where the opacity of the edge indicates the similarity score (weaker links are more transparent). Visually, the matching defines a clear alignment of the two graphs: in the bottom left and centre we have the Mathematics-related concepts; at the right we have Biology-related concepts; and at the top left we have Economics-related concepts.

### A.4 Visualisation of generated ontologies

### A.4.1 Wikipedia

We include some generated outputs for Wikipedia here. Since the full generated output is too large to visualise, we plot subgraphs of the output instead. We sample the subgraphs by the following method:

1. Pick a random node in the generated graph.
2. Get the induced subgraph by the 1-hop neighbourhood of the chosen node.
3. Include the shortest path from the root "Main topic classifications" to the chosen node if such path exists.
4. Repeat from step 1 if the subgraph has more than 30 nodes or less than 5 nodes.

We apply the filtering step (step 4) as subgraphs with too many nodes are difficult to inspect manually, and those with too few are uninformative. For Hearst, we choose the filtering upper bound to be 50 nodes as we fail to find subgraphs smaller than 30 nodes quickly. We additionally colour each edge **black** if it occurs literally in the training graph, blue if it occurs literally in the test graph, and red otherwise.

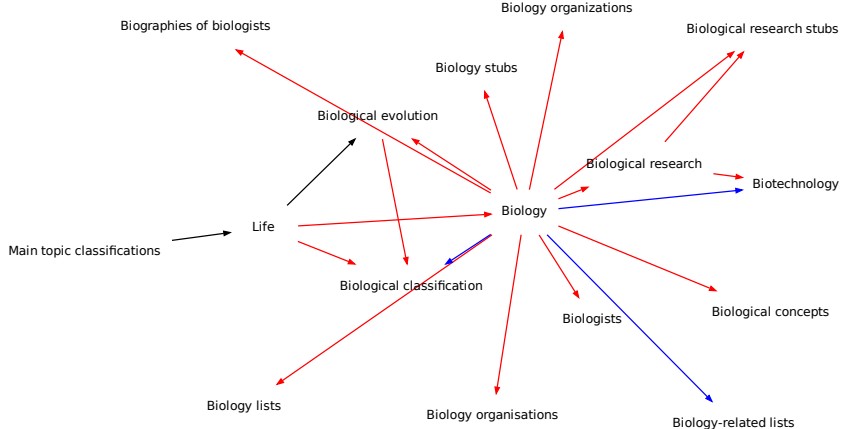

(a) Biology

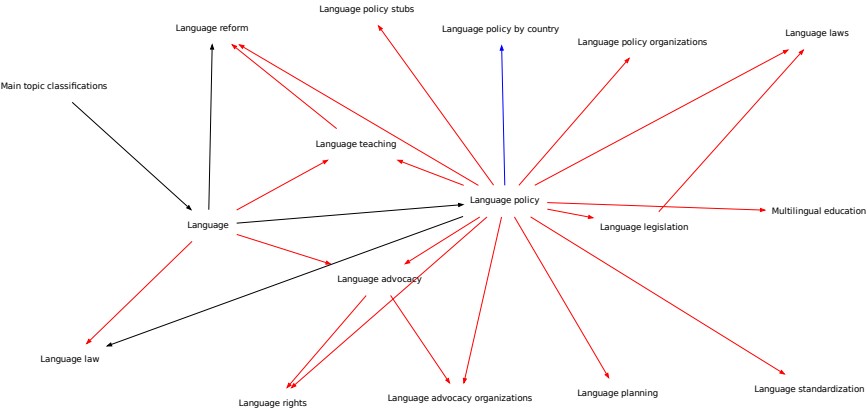

(b) Language policy

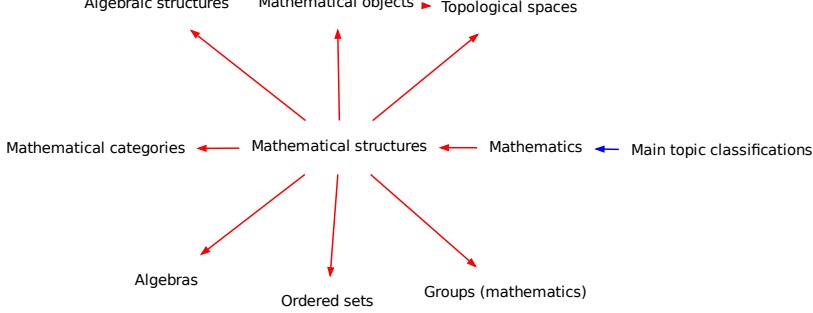

(c) Mathematical structures

Figure 11: Sub-ontologies for Wikipedia generated by OLLM, centred on various topics.

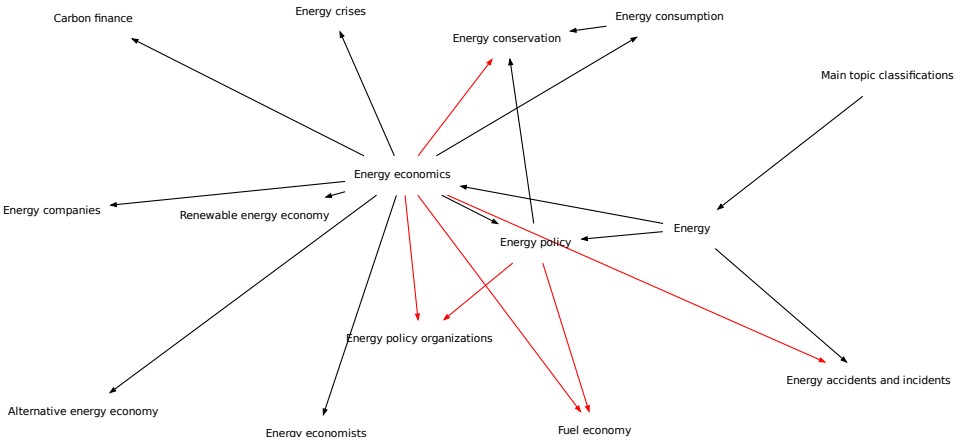

(a) Energy economics

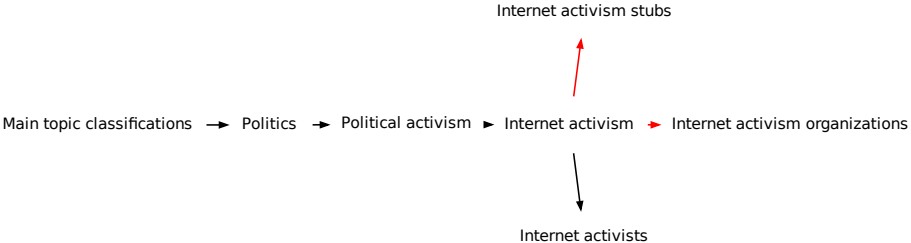

(b) Internet activism

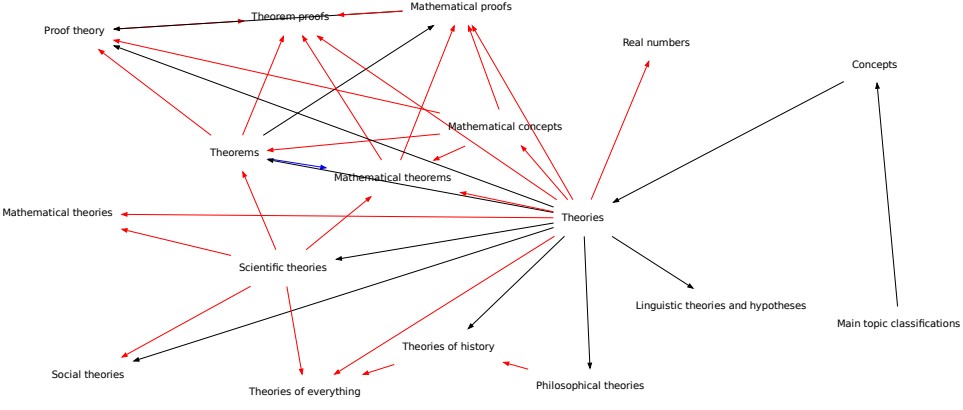

(c) Theories

Figure 12: Sub-ontologies for Wikipedia generated by Finetune, centred on various topics.

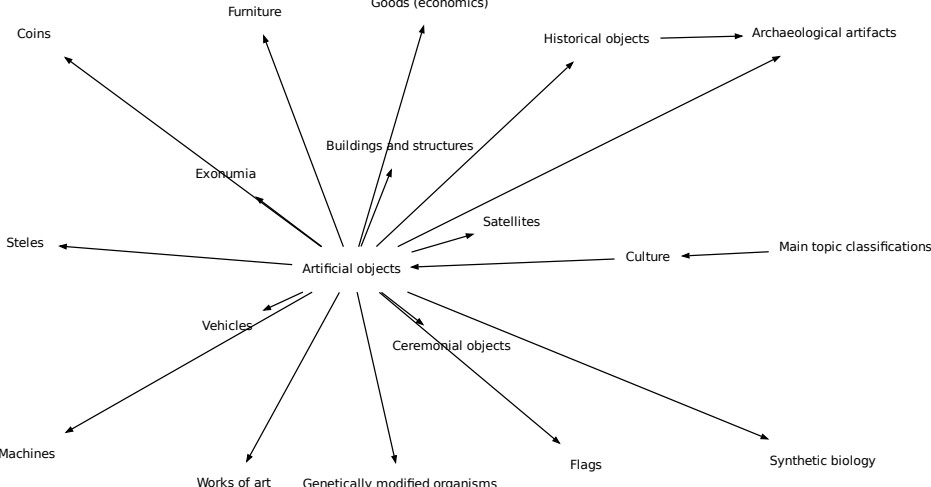

(a) Aritificial objects

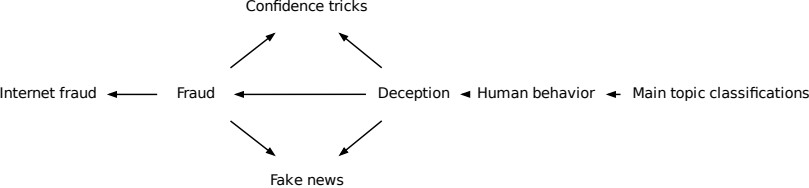

(b) Fraud

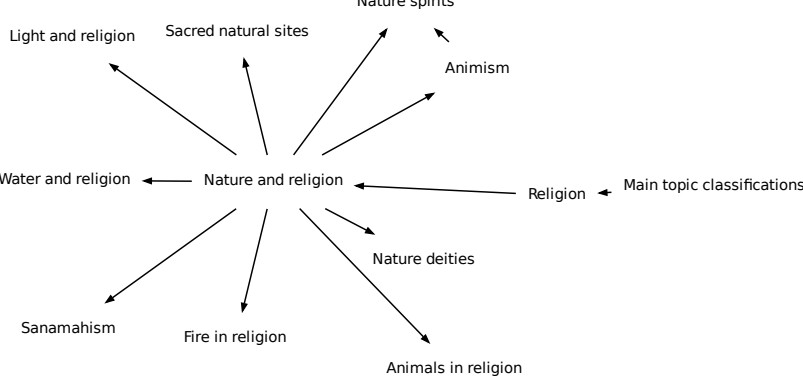

(c) Nature and religion

Figure 13: Sub-ontologies for Wikipedia generated by Memorisation, centred on various topics.

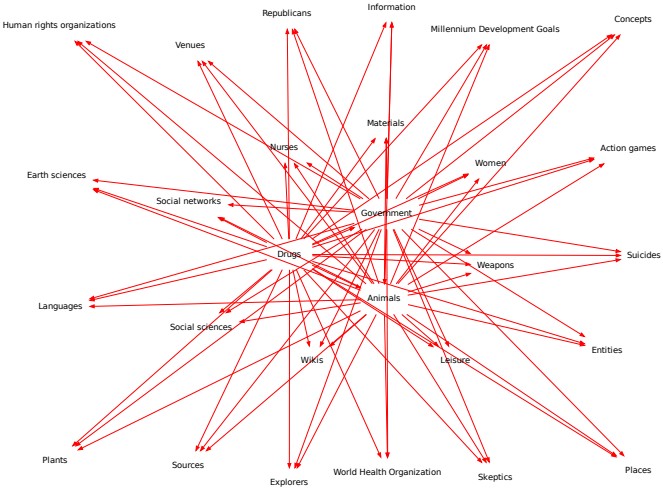

(a) Drugs

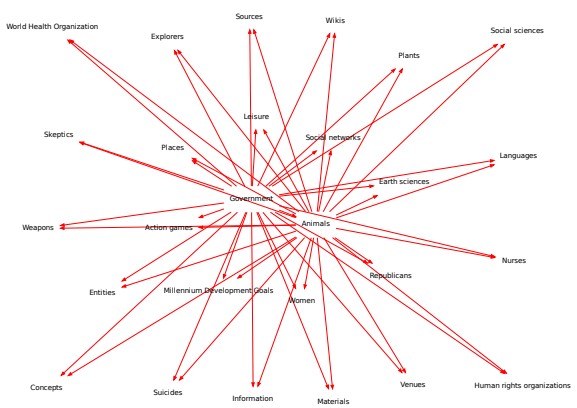

(b) Government

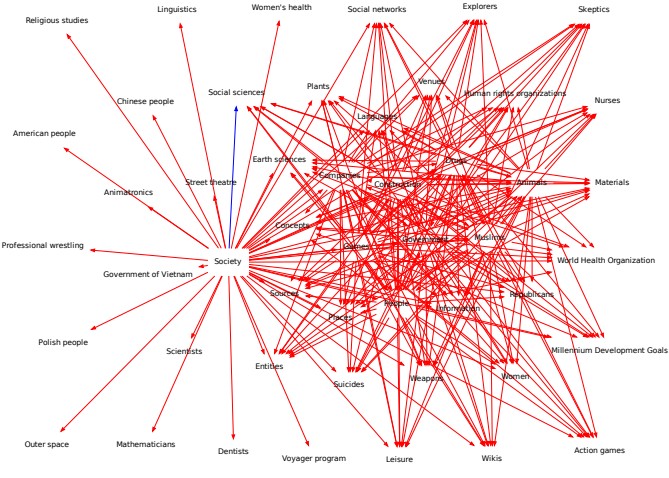

(c) Society

Figure 14: Sub-ontologies for Wikipedia generated by Hearst, centred on various topics.

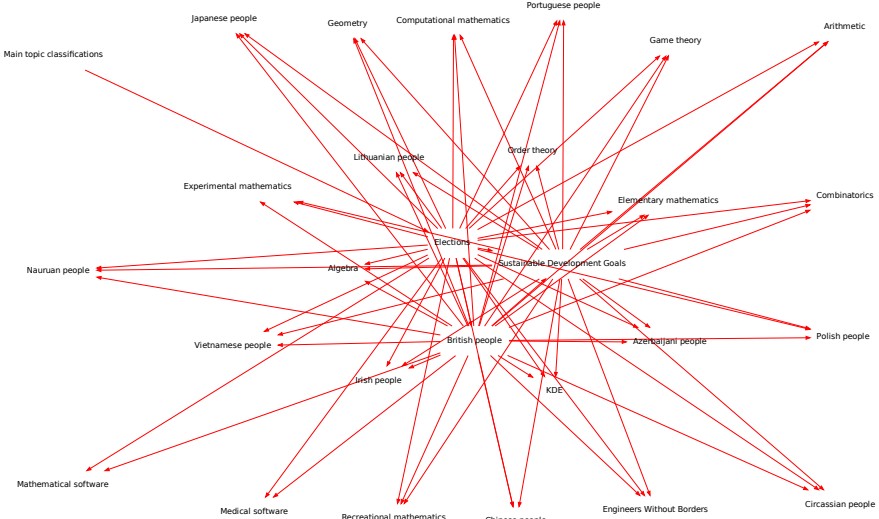

(a) Elections

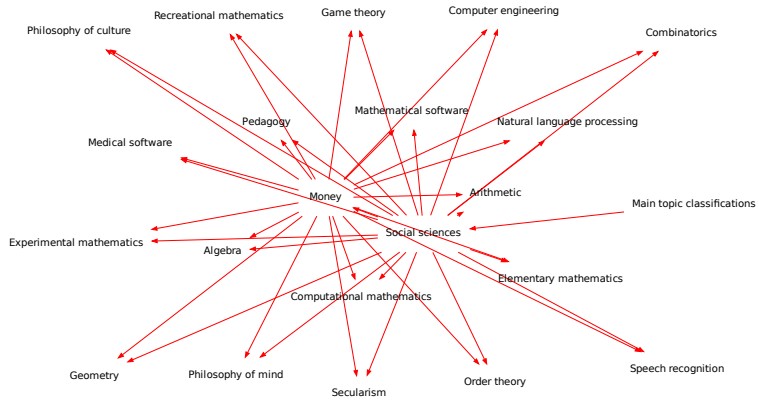

(b) Money

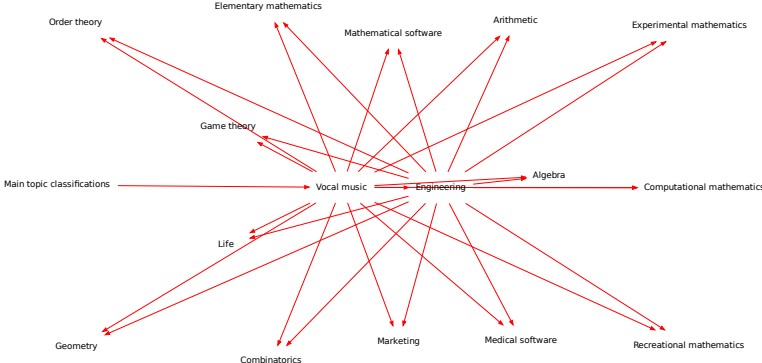

(c) Vocal music

Figure 15: Sub-ontologies for Wikipedia generated by REBEL, centred on various topics.

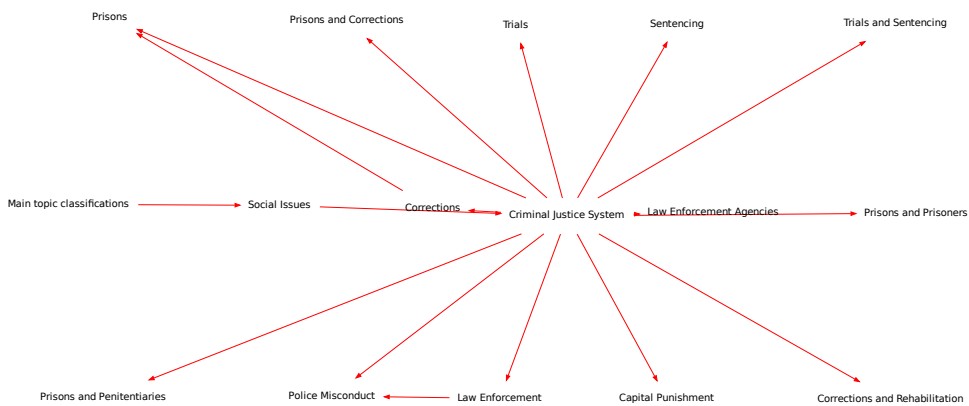

(a) Criminal Justice System

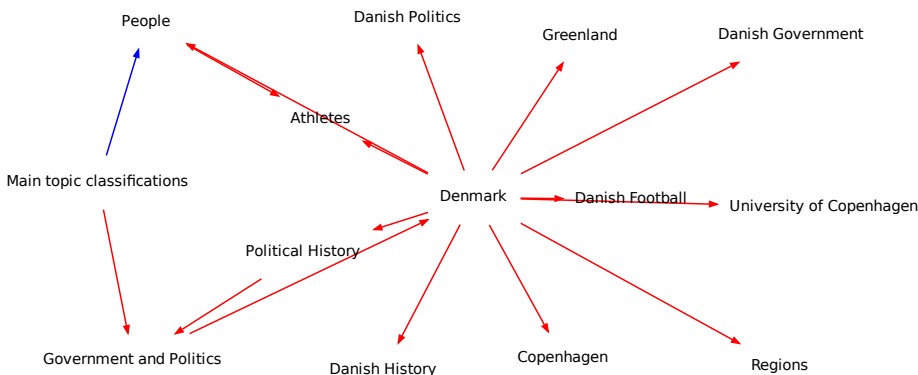

(b) Denmark

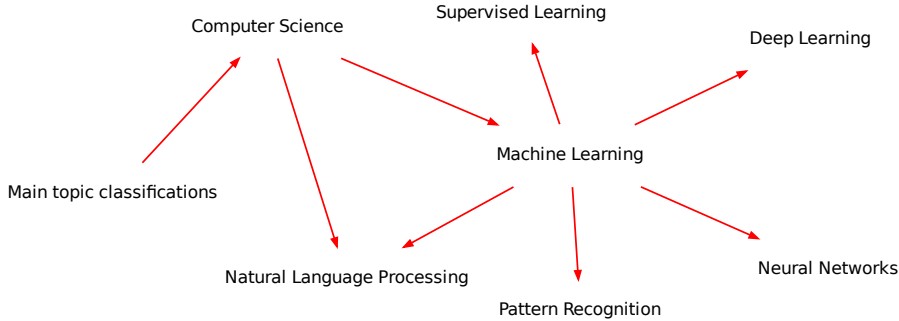

(c) Machine Learning

Figure 16: Sub-ontologies for Wikipedia generated by Zero-shot, centred on various topics.

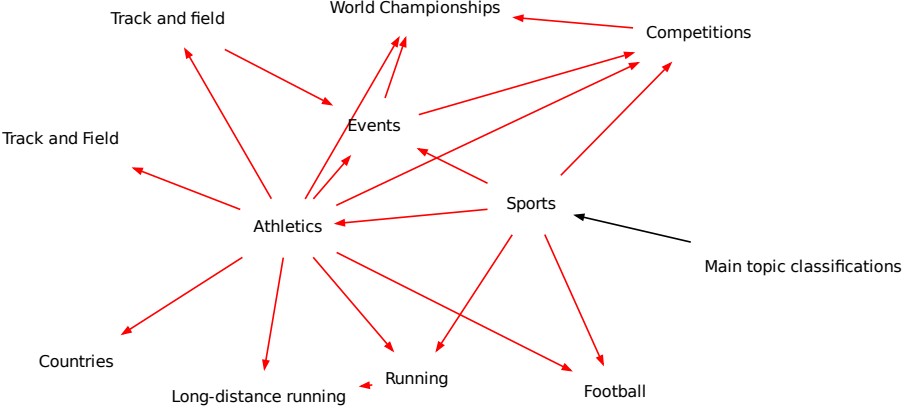

(a) Athletics

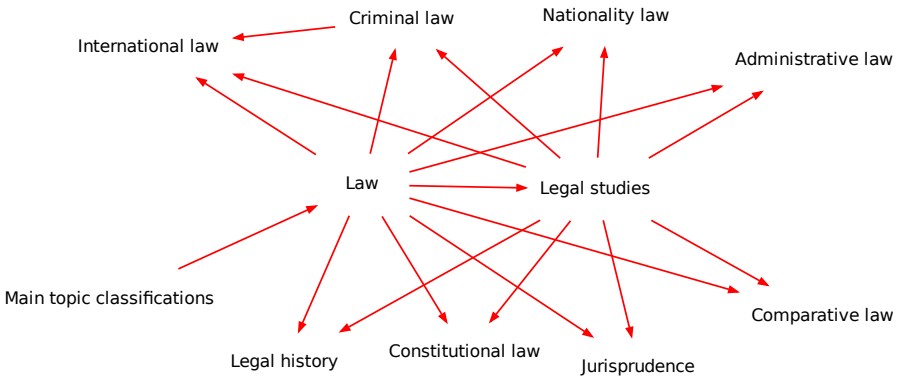

(b) Legal studies

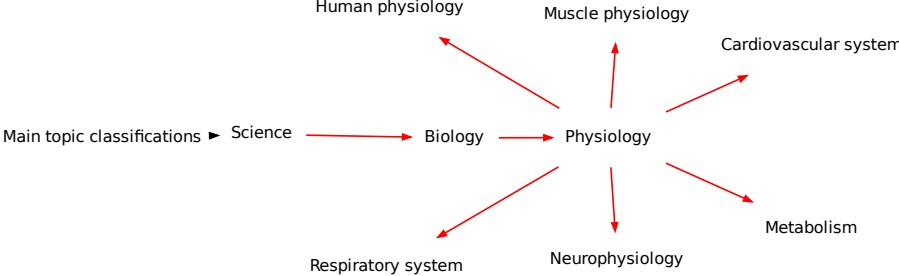

(c) Physiology

Figure 17: Sub-ontologies for Wikipedia generated by One-shot, centred on various topics.

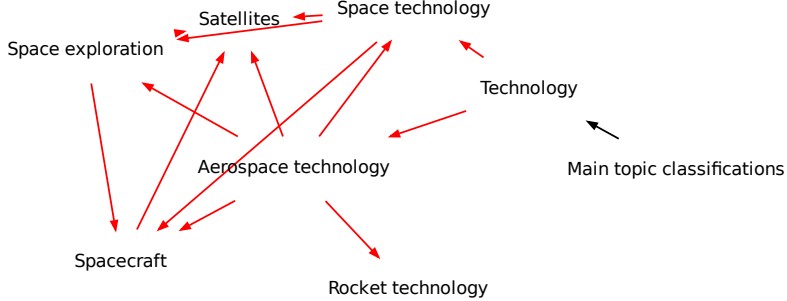

(a) Aerospace technology

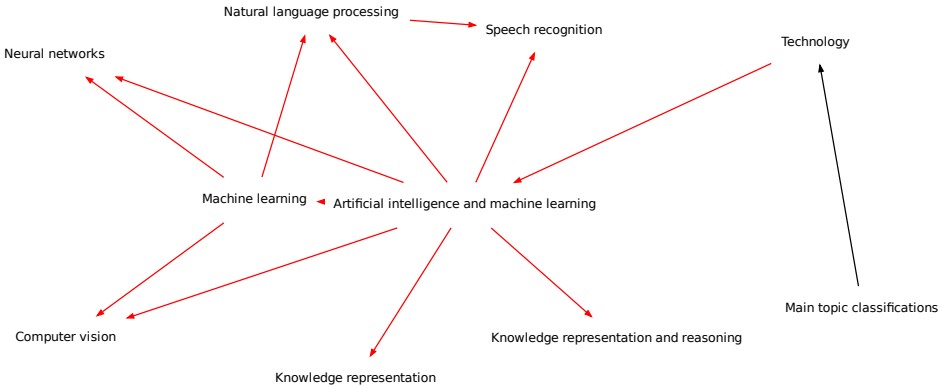

(b) Artificial intelligence and machine learning

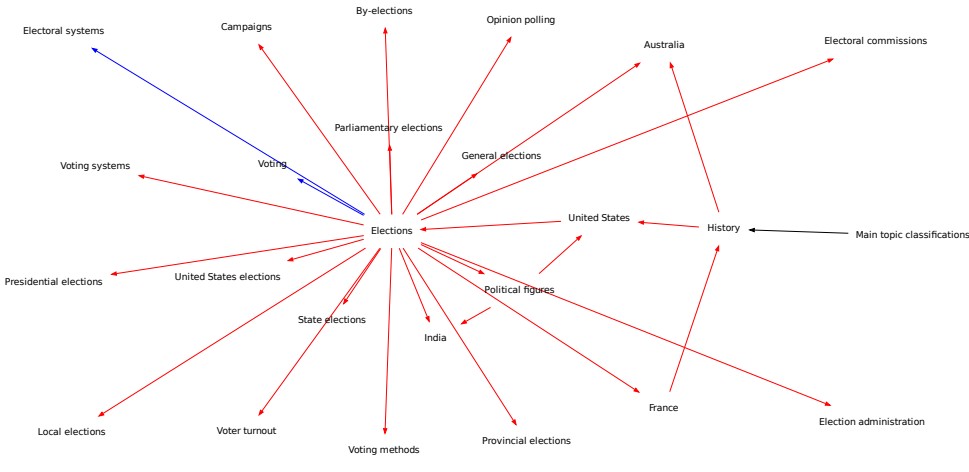

(c) Elections

Figure 18: Sub-ontologies for Wikipedia generated by Three-shot, centred on various topics.

## A.4.2 arXiv

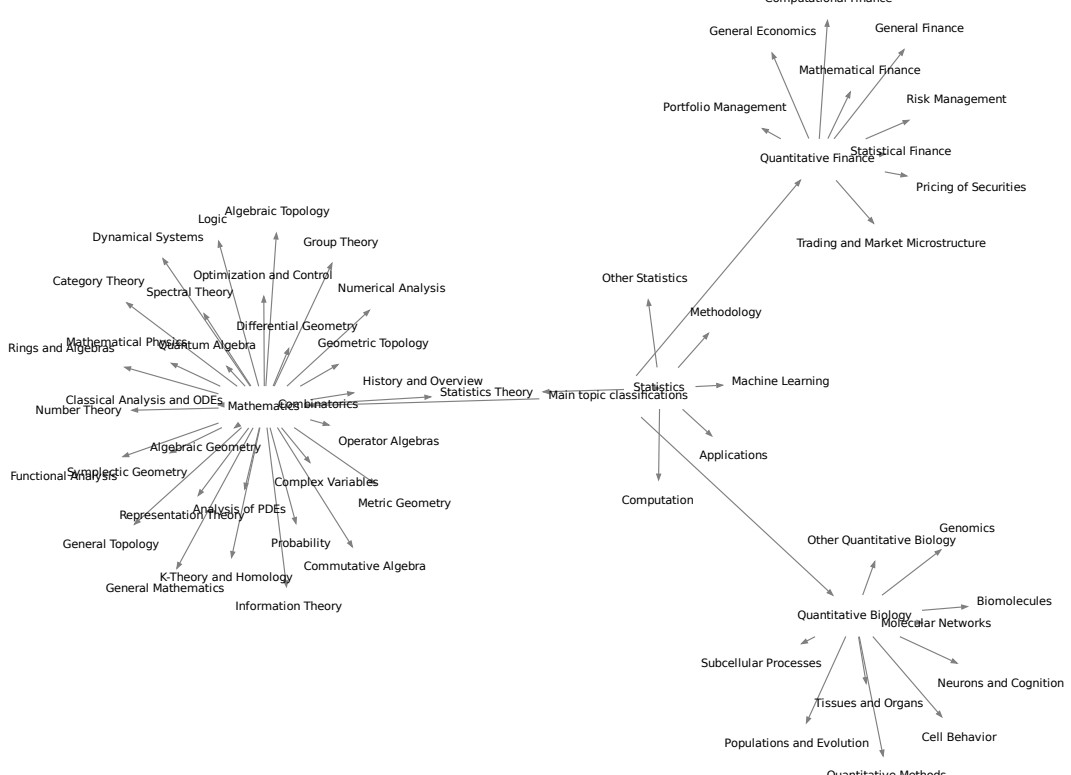

Figure 19: Ground truth test split ontology for arXiv

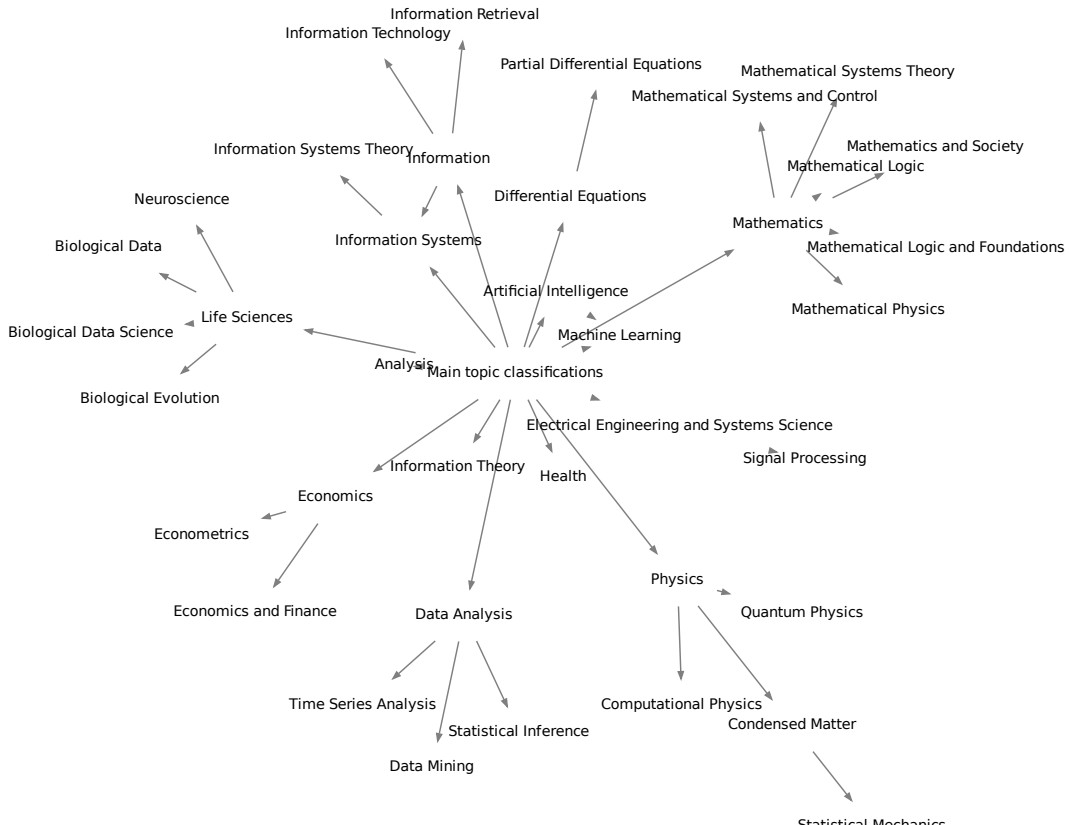

Figure 20: Ontology for arXiv generated by OLLM

Figure 21: Ontology for arXiv generated by Finetune

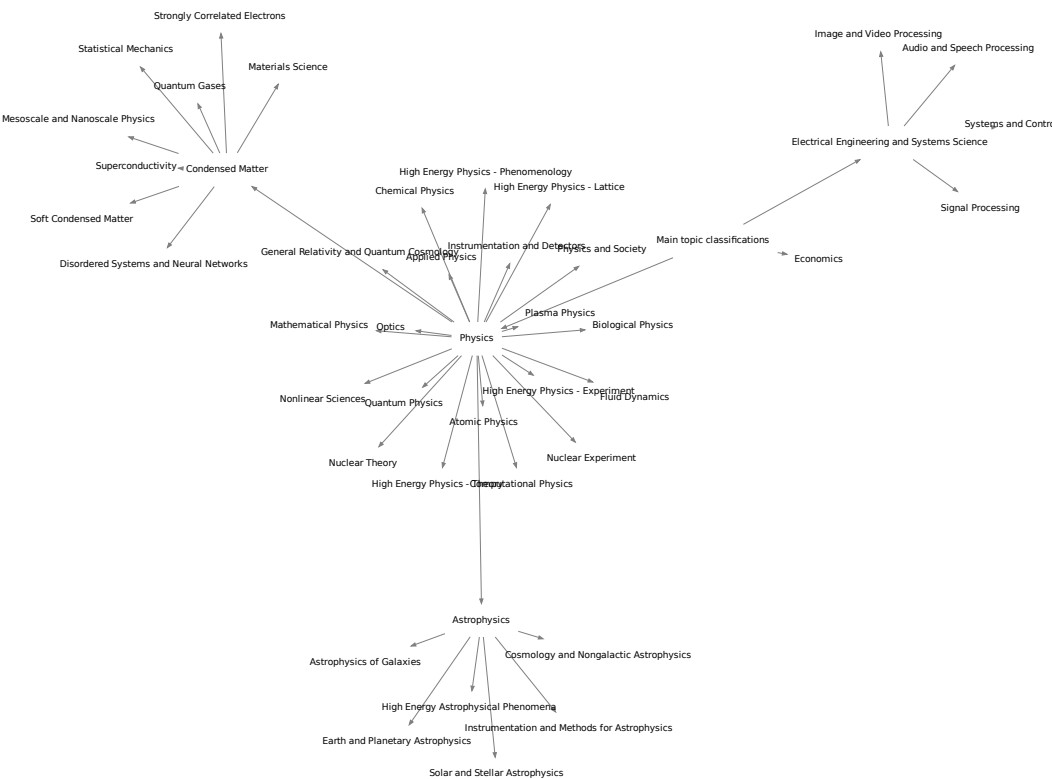

Figure 22: Ontology for arXiv generated by Memorisation

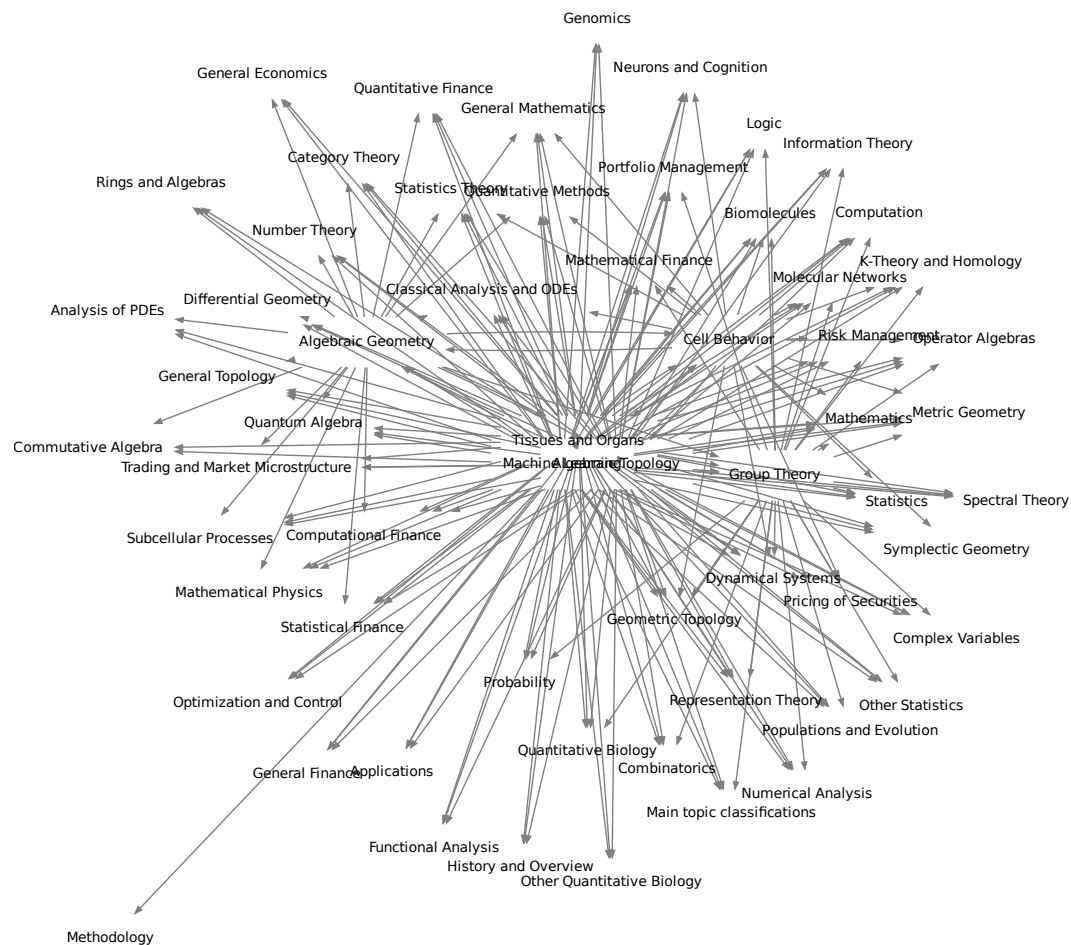

Figure 23: Ontology for arXiv generated by Hearst

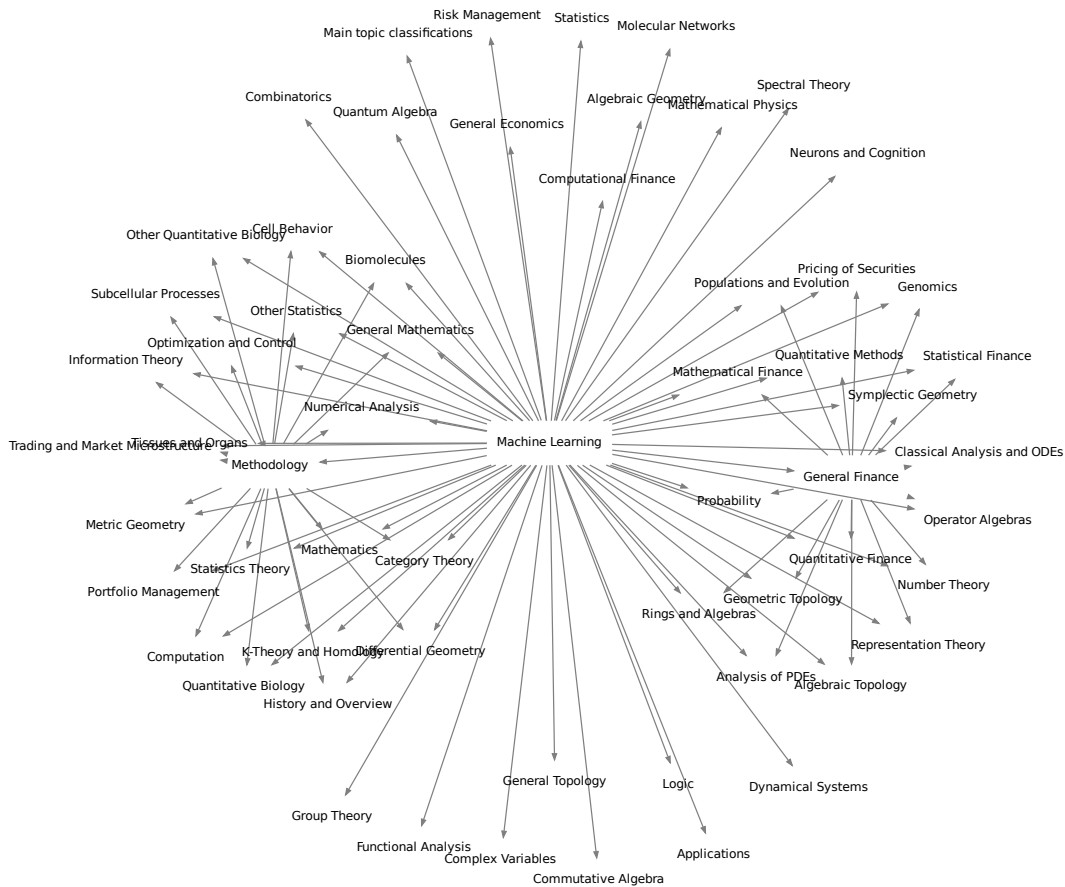

Figure 24: Ontology for arXiv generated by REBEL

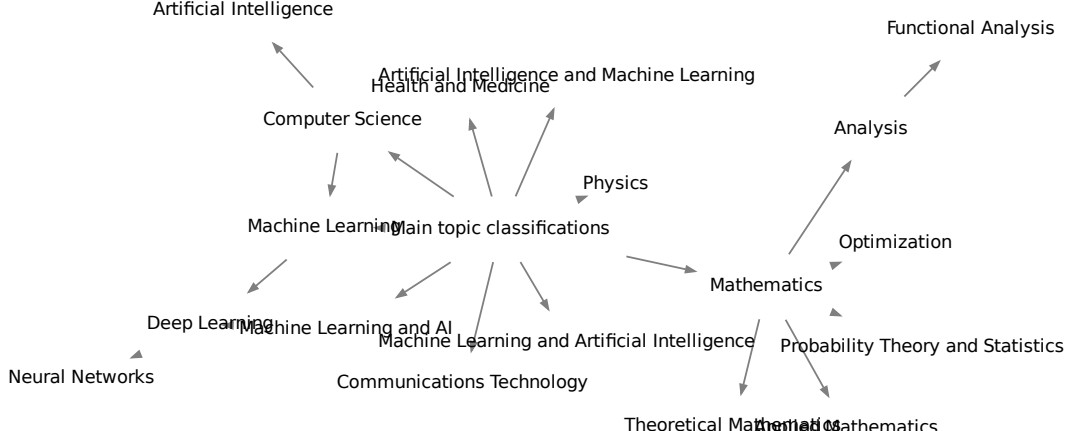

Figure 25: Ontology for arXiv generated by Zero-shot

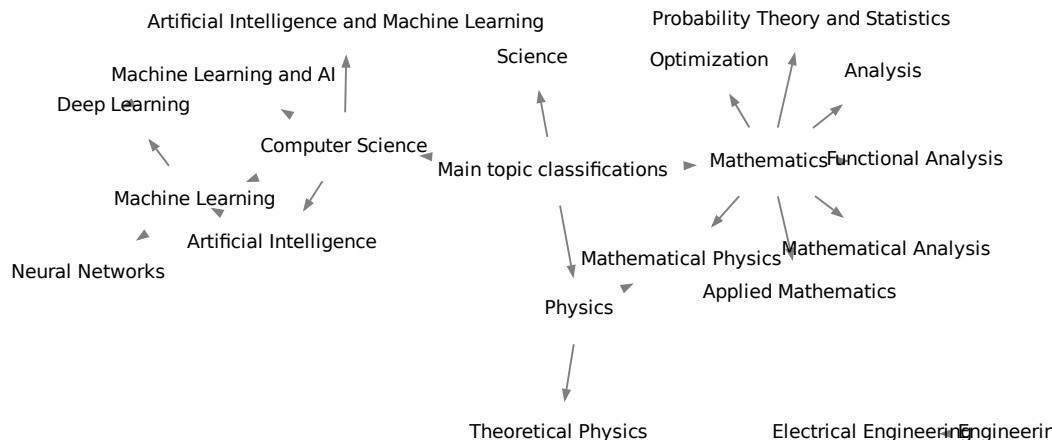

Figure 26: Ontology for arXiv generated by One-shot

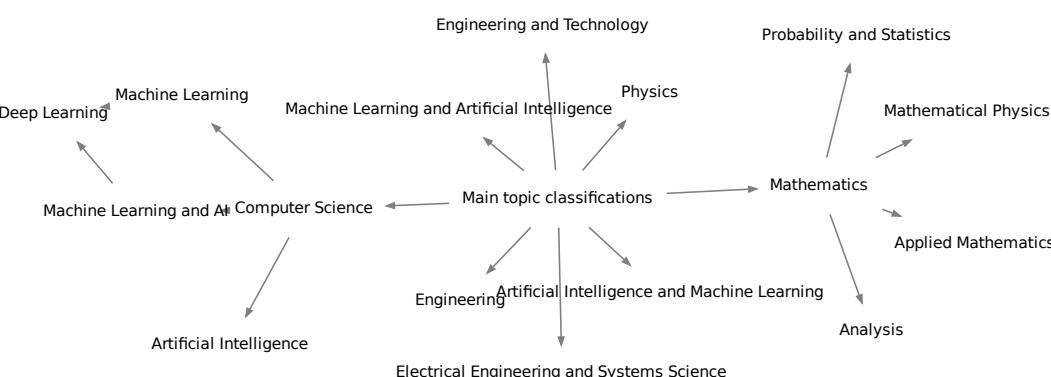

Figure 27: Ontology for arXiv generated by Three-shot

