# OpenReview forum: "End-to-End Ontology Learning with Large Language Models"
_NeurIPS.cc/2024/Conference — NeurIPS 2024 poster_

### Official Review · Reviewer_Yksm · 2024-07-07

**Soundness:** 2
**Presentation:** 3
**Contribution:** 2
**Rating:** 7
**Confidence:** 4

**Summary:**

This paper proposes a new methodology of Ontology Learning using Large Language Models and ontology paths which is called OLLM. After introducing two ontology-datasets for Wikipedia and ArXiv, they train an LLM using the titles, summaries, and ontology paths to allow for sub-graph generation. The method is additionally improved using post-processing techniques for summing and pruning the subgraphs with the purpose of building a complete ontology. New metrics are also presented for facilitating the comparison of the generated ontology with the ground truth and other baseline techniques.

**Strengths:**

•	This work presents a novel technique leveraging the power of LLMs and ontology paths for ontology learning.

•	Two different ontology-datasets (Wikipedia and ArXiv) are constructed and used for the experiments.

•	New metrics for ontology evaluation are proposed based on semantic similarity.

•	OLLM seems to outperform most of the baseline methods in the experiments.

•	Overall, the paper is well-written.

**Weaknesses:**

•	While the OLLM results are better than most of the baseline methods that is not the case using the motif distance metric (Wikipedia: Finetune = 0.05 while OLLM = 0.080, Arxiv: Memorisation = 0.037 while OLLM = 0.097). That is not well explained in the paper.

•	Performance seems to drop significantly when using a smaller dataset such as arXiv.

•	It is unclear whether OLLM can generalize well to different domains without having an initial ontology for extracting the paths.

**Questions:**

•	Usually training sets consist of the biggest part of a dataset than validation and test sets. Why did you choose such small portion of the datasets for training? Could that be one reason for the initial poor generalisation?

•	It is also unclear how much the overlapping parts of training/validation/test sets contribute to the final result.

**Limitations:**

The main limitation, to my opinion, is that OLLM cannot generate a complete ontology for a different domain without already having an initial ontology for extracting the paths first. This means that there is still the need of reusing or constructing some part of an ontology.

---

> ### Author Rebuttal · Authors · 2024-08-06
>
> We thank the reviewer for their detailed feedback.
>
> > While the OLLM results are better than most of the baseline methods that is not the case using the motif distance metric (Wikipedia: Finetune = 0.05 while OLLM = 0.080, Arxiv: Memorisation = 0.037 while OLLM = 0.097). That is not well explained in the paper.
>
> We thank the reviewer for raising this point. As the data splits are constructed symmetrically (i.e., the train, validation, and test splits have the same distribution), we expect all data splits to have similar structures. This implies that Motif Distance is biased towards overfitting, since the structure of the train and test splits tend to be similar, even though the represented concepts are different. This explains why methods that tend to overfit (i.e., Memorisation and Finetune) achieve the best scores. We will include this discussion in Section 5.3 in the final version.
>
> > Performance seems to drop significantly when using a smaller dataset such as arXiv.
>
> We would like to emphasise that the two experiments are designed to answer different questions and thus the results are not suitable for direct comparison. We use Wikipedia to test the model’s ability to generalise in-domain, whereas the arXiv experiment tests the transferability using only a small number of training examples. The arXiv task also appears to be more difficult in general, as indicated by the worse performance of the baselines.
>
> > It is unclear whether OLLM can generalize well to different domains without having an initial ontology for extracting the paths.
>
> We agree that training examples are needed to give the best performance. Ontologies come in different “styles” (e.g., the granularity of concepts) which is difficult to specify without training examples. However, we argue that our end-to-end modelling approach is useful in other domains:
>
> 1. In the arXiv experiment, we only used 2000 finetuning examples to transfer from Wikipedia to arXiv and achieved good performance. This is a modest cost if one were to apply our model to a new domain.
> 2. Even without training examples, our 0/1/3-shot prompting method outperforms existing baselines (including the new ablation methods, see general response).
>
> > Usually training sets consist of the biggest part of a dataset than validation and test sets. Why did you choose such small portion of the datasets for training? Could that be one reason for the initial poor generalisation?
>
> We deliberately chose to use a larger test split as we wanted to ensure that sufficiently many concepts and relations were unseen in the training split. This tests the model’s ability to generalise in-domain. Using a larger split for evaluation also helps to get more reliable metrics, as we did not do repeated runs.
>
> As described in Section 5.1, the fundamental cause of poor generalisation of direct finetuning is due to the imbalanced sampling of high and low-level concepts. This results in the model learning high-level concepts much faster, resulting in overfitting high-level concepts while underfitting low-level ones. Increasing the training set size does not solve such issue.
>
> > It is also unclear how much the overlapping parts of training/validation/test sets contribute to the final result.
>
> It is inevitable and desirable that there is an overlap between the data splits. We attempt to quantify the contribution of such overlap to the final result by introducing the “Memorisation” baseline method, which demonstrates the quality of resultant ontology if one were to blindly exploit such overlap. The fact that Memorisation performs poorly suggests that the overlaps do not contribute significantly to the final result.
>
> ---
>
> Would the reviewer please consider raising the score if we have addressed their concerns?

---

> > ### Comment · Reviewer_Yksm · 2024-08-10
> >
> > Thank you for the detailed explanations and additional results. Since my concerns were addressed, I am increasing my score.

---

### Official Review · Reviewer_3YVJ · 2024-07-11

**Soundness:** 3
**Presentation:** 4
**Contribution:** 3
**Rating:** 7
**Confidence:** 4

**Summary:**

The paper is well-written, addresses a valuable real-world problem, and proposes a simple, intuitive, and effective method. The experiments are rigorously designed with comprehensive evaluation metrics. Overall, this is a high-quality work with notable contributions.

However, there are concerns about whether the improved performance is due to end-to-end modeling or LLM capabilities, the assumption that each document is associated with at least one concept, the consistency of the generated ontology, and the post-processing procedures.

**Strengths:**

- S1: The paper is well-written and easy to follow
- S2: The investigated ontology learning problem is of great value for real-world applications
- S3: The proposed method is simple, intuitive, yet problem-oriented, novel, and effective
- S4: The experiments are well-designed, in particular, the baselines and the data-splitting strategy are rigorously
- S5: The set of evaluation metrics is comprehensive and covers various aspects of performance

**Weaknesses:**

- W1: The paper's main claim is that end-to-end modeling is better than pipelined methods. However, it is unclear whether the improved performance is credited to the end-to-end modeling approach or the capabilities of LLMs. Although I am convinced that end-to-end + LLM is a good solution, I am curious if applying LLMs for the subtasks in the pipeline will work even better, especially considering LLMs are good at decomposing problems and solving them step-by-step.
- W2: The authors assumed each document is associated with at least one concept in the ground truth ontology. However, in real-world applications, it would be very possible that the whole document does not contain any relevant edges or even concepts. Can the proposed OLLM say no if a random irrelevant document is given?
- W3: Consistency is a crucial property of ontologies, which is the prerequisite to run most logical reasoners. Is the generated ontology guaranteed to be consistent?
- W4: I am concerned about the post-processing procedure. Are there any entity (concept) resolution mechanisms involved to merge semantically equivalent concepts? For example, if “English” and “english” are both there, are then treated as the same concept? And for “English Language” and “The Language Named English”, are they merged?
- W5: Why using 3 prompts is generally worse than 1-shot?

**Questions:**

Please refer to W1-W5.

**Limitations:**

Yes.

---

> ### Author Rebuttal · Authors · 2024-08-06
>
> We thank the reviewer for their detailed feedback.
>
> > W1: The paper's main claim is that end-to-end modelling is better than pipelined methods. However, it is unclear whether the improved performance is credited to the end-to-end modelling approach or the capabilities of LLMs. Although I am convinced that end-to-end + LLM is a good solution, I am curious if applying LLMs for the subtasks in the pipeline will work even better, especially considering LLMs are good at decomposing problems and solving them step-by-step.
>
> We thank the reviewer for their suggestion and have run additional ablations based on applying LLMs for subtasks. The full experiment setup can be found in the general response. Here, we summarise the results:
> 1. The concept discovery + link prediction-based approach (studied in LLMs4OL [2]) suffers from serious scalability bottlenecks. In particular, it requires $O(n^2)$ inferences for link prediction, which is impractical for large models and/or large number of concepts. This is the reason why this baseline was not used originally. For the ablation, we made ad-hoc fixes to the method to make it manageable for our large problem sizes (see general response).
> 2. The results show that it is generally worse than our zero-shot prompting method based on subgraph modelling.
>
> We believe this is further evidence that our subgraph modelling method is a good solution.
>
> > W2: The authors assumed each document is associated with at least one concept in the ground truth ontology. However, in real-world applications, it would be very possible that the whole document does not contain any relevant edges or even concepts. Can the proposed OLLM say no if a random irrelevant document is given?
>
> We do not view this assumption as a fundamental limit to OLLM. One simple modification we can make is to include empty targets in the training set such that the model learns to output an empty graph if the document does not contain any relevant concepts. We made this assumption as it applies to our datasets and simplifies the implementation.
>
> > W3: Consistency is a crucial property of ontologies, which is the prerequisite to run most logical reasoners. Is the generated ontology guaranteed to be consistent?
>
> Ensuring consistency is a challenge for many OL methods and is not specific to OLLM. There exist generic post-processing methods that can guarantee consistency [6]. We performed further analysis of the generated output by OLLM and found that they are near-consistent: the output ontology for Wikipedia consists only of 97 simple cycles and no cycles were found in arXiv. Using the greedy algorithm of repeatedly removing the edge that breaks the most simple cycles (a heuristic to the smallest set of edges whose removal makes the graph acyclic), we prune all such cycles and make the ontology consistent by removing just 26 out of 10414 edges. This is quite surprising considering we did not explicitly optimise our model to satisfy consistency.
>
>
> > W4: I am concerned about the post-processing procedure. Are there any entity (concept) resolution mechanisms involved to merge semantically equivalent concepts? For example, if “English” and “english” are both there, are then treated as the same concept? And for “English Language” and “The Language Named English”, are they merged?
>
> We do not perform any concept-merging in our post-processing steps. While we agree that applying such normalisation strategies can produce better ontologies, we do not think it contributes to answering the research question of “Does OLLM produce better ontologies?”. Instead, we decided to restrain the post-processing steps to be simplistic and minimal without relying on many heuristics. This allows us to attribute any differences in output quality to the method itself without being confounded by interactions with the post-processing procedure.
>
>
> > W5: Why using 3 prompts is generally worse than 1-shot?
>
> We thank the reviewer for raising such concerns. In response, we did a short investigation focusing on arXiv (as that’s where 3-shot seemed to perform worse than 1-shot). We found that 3-shot appears to generate unnecessarily long responses, which we hypothesise leads to noisier outputs. Across the training set, the ground truth subgraph targets contain a median of 4.0 concepts (IQR: 4.0). In contrast, 1-shot predicts a median of 9.0 (IQR: 8.0) and 3-shot predicts a median of 11.0 (IQR: 9.0). We see that giving more examples tends to result in longer responses, which might be an inherent bias of Mistral 7B.
>
> We note that the performance gap between 1-shot and 3-shot is generally insignificant so we do not believe the claim “3-shot is generally worse than 1-shot” is fully justified.
>
> ---
>
> Would the reviewer please consider raising the score if we have addressed their concerns?
>
> [6]: Sun, Jiankai, et al. "Breaking cycles in noisy hierarchies." Proceedings of the 2017 ACM on Web Science Conference. 2017.

---

> > ### Author Response · Authors · 2024-08-12
> >
> > Again, we thank the reviewer for their constructive feedback. As we are near the end of the discussion period, we would like to confirm if our answers are comprehensive and satisfactory. Should any new questions arise, we would be happy to answer them.

---

> > ### Comment · Reviewer_3YVJ · 2024-08-13
> >
> > Thanks for the rebuttal. It solved most of my concerns. I will keep my evaluation.

---

### Official Review · Reviewer_cxnM · 2024-07-11

**Soundness:** 3
**Presentation:** 3
**Contribution:** 1
**Rating:** 4
**Confidence:** 4

**Summary:**

The paper introduces a novel method called OLLM (Ontology Learning with Large Models) for automating the construction of ontologies using large language models (LLMs). Ontologies are crucial for structuring domain-specific knowledge, but their manual construction is labor-intensive. The authors aim to address the limitations of partial ontology learning approaches by proposing an end-to-end method to build ontologies from scratch.

This paper demonstrates OLLM's effectiveness through quantitative and qualitative results on Wikipedia data, showing that it outperforms subtask composition methods in producing more semantically accurate ontologies while maintaining structural integrity. Additionally, OLLM's adaptability to new domains like arXiv is showcased, requiring only a small number of training examples for effective adaptation.

**Strengths:**

1) The authors propose a custom regularizer that reweights concepts based on their frequency of occurrence, which helps to mitigate overfitting on common concepts and enhances the model's ability to generalize to new, unseen data.
2) The paper introduces a novel suite of evaluation metrics that use deep learning techniques to measure semantic and structural similarity between ontologies. These metrics provide a more robust and nuanced assessment compared to traditional methods.
3) The authors demonstrate the effectiveness of OLLM through comprehensive experiments on Wikipedia and arXiv datasets. The results show that OLLM outperforms existing subtask-based methods, indicating its practical utility.

**Weaknesses:**

1) The paper focuses on building ontologies with concepts and taxonomic relations, which is only a part of the full spectrum of ontology components. A comprehensive ontology includes not only hierarchical relationships but also properties, semantic relations, constraints, and other elements that provide a richer and more explicit semantic content. The paper does not address how these additional aspects are captured and integrated into the learned ontologies.
2) The paper uses Wikipedia categories and arXiv taxonomy as the basis for ontology learning. However, these structures are more accurately described as taxonomies or folsonomies rather than full-fledged ontologies, which can lead to confusion about the capabilities and outputs of the proposed method.
3) The paper does not provide a clear and detailed explanation of the end-to-end ontology learning model's training process. Key aspects such as the construction of the training dataset, filtering of irrelevant categories, and the specific model architecture and training regimen are not thoroughly described. For example, it is not clear how the model deals with Wikipedia categories that have little semantic content but are used for page management.
4) The paper does not include a comparative analysis with other ontology learning methods, such as the LLMs4OL (LLMs4OL: Large Language Models for Ontology Learning, https://arxiv.org/abs/2307.16648.). Without such comparisons, it is difficult to assess the relative advantages and disadvantages of the proposed method in terms of effectiveness, efficiency, and scalability.

**Questions:**

See weaknesses.

**Limitations:**

See weaknesses.

---

> ### Author Rebuttal · Authors · 2024-08-06
>
> We thank the reviewer for their detailed feedback.
>
> > The paper focuses on building ontologies with concepts and taxonomic relations, which is only a part of the full spectrum of ontology components. A comprehensive ontology includes not only hierarchical relationships but also properties, semantic relations, constraints, and other elements that provide a richer and more explicit semantic content. The paper does not address how these additional aspects are captured and integrated into the learned ontologies.
> > The paper uses Wikipedia categories and arXiv taxonomy as the basis for ontology learning. However, these structures are more accurately described as taxonomies or folsonomies rather than full-fledged ontologies, which can lead to confusion about the capabilities and outputs of the proposed method.
>
> We agree that our method only focuses on the basic components of an ontology, and such structures are perhaps better described as taxonomies. We thank the reviewer for raising this point and will refrain from phrases like “solving the full task of building an ontology” (L7) in the final version of the paper for better clarity.
>
> The main contribution of this paper is to use LLMs to construct basic ontologies end-to-end as one task, and doing so yields more accurate concepts and taxonomic relations. We view more complex aspects of ontologies as extensions and thus beyond the scope of this paper. We hope the reviewer can view our achievements in comparison to existing methods for modelling concepts and taxonomic relations, and agree that improving the quality of the fundamental components of an ontology is a sufficient contribution already.
>
> > The paper does not provide a clear and detailed explanation of the end-to-end ontology learning model's training process. Key aspects such as the construction of the training dataset, filtering of irrelevant categories, and the specific model architecture and training regimen are not thoroughly described.
>
> In Section 4.1, we give a detailed description of how we construct our datasets, including the process of obtaining the raw data and how we construct the data splits. Section 3.1 describes how we obtain document-subgraph pairs from the source data. We did not perform any additional filtering of categories, hence not such mention in the paper. Due to space limitations, we did not include all implementation details. We will consider rewording relevant parts of the paper for better clarity and include more details in the appendix.
>
> Similarly, the specific model architecture and training details are broadly mentioned in the main text. As described in Section 5.1, OLLM is a LoRA finetune of Mistral 7B which has its architecture described in [5]. We also included a detailed description of all the LoRA and training hyperparameters in Appendix A.1.1 to aid reproducibility.
>
> We also share our dataset, model and code (including dataset construction and training), which hopefully resolves all ambiguities.
>
> > For example, it is not clear how the model deals with Wikipedia categories that have little semantic content but are used for page management.
>
> We did not apply any filtering to the source data, hence our model will learn to construct and organise every concept in the dataset as part of at subgraph modelling stage. We decided to not apply any filtering to the source data to minimise external bias on how the data “should” look like. We note that it is often not clear-cut whether a Wikipedia category is just for page management: for example, the Wikipedia categories of the form “Lists of [subject]” refer to the special type of articles where the main body is a bullet point/table listing of the subject, which is a useful concept in the Wikipedia domain.
>
> > The paper does not include a comparative analysis with other ontology learning methods, such as the LLMs4OL (LLMs4OL: Large Language Models for Ontology Learning, https://arxiv.org/abs/2307.16648.).
>
> We thank the reviewer for their suggestion and have run additional ablations based on LLMs4OL. The full experiment setup can be found in the general response. Here, we summarise the results:
>
> 1. The link prediction-based approach studied in LLMs4OL suffers from serious scalability bottlenecks. In particular, it requires $O(n^2)$ inferences for link prediction which is impractical for large models and/or large number of concepts. This is the reason why this baseline was not used originally. For the ablation, we made ad-hoc fixes to the method to make it manageable for our large problem sizes (see general response).
> 2. The results show that it is generally worse than our zero-shot prompting method based on subgraph modelling.
> We will include the above discussion points regarding LLMs4OL in the final version of the paper.
>
> ---
>
> Would the reviewer please consider raising the score if we have addressed their concerns?
>
> [5]: Jiang, Albert Q., et al. "Mistral 7B."

---

> > ### Author Response · Authors · 2024-08-12
> >
> > Again, we thank the reviewer for their constructive feedback. As we are near the end of the discussion period, we would like to confirm if our answers are comprehensive and satisfactory. Should any new questions arise, we would be happy to answer them.

---

### Official Review · Reviewer_Qo2b · 2024-07-12

**Soundness:** 2
**Presentation:** 3
**Contribution:** 2
**Rating:** 4
**Confidence:** 4

**Summary:**

The paper aims to address the challenge of constructing ontologies, which traditionally require substantial manual effort. Ontologies are structured representations of domain knowledge used for automatic machine processing. Previous methods for ontology learning (OL) using large language models (LLMs) focused on solving individual subtasks, but this approach failed to capture the interactions between these subtasks. Experimental results on Wikipedia demonstrate that the authors' proposed approach, called OLLM, outperforms traditional subtask composition methods, producing more semantically accurate ontologies while maintaining structural integrity. The model also shows effective adaptation to new domains, such as arXiv, with only a small number of training examples, indicating its scalability and domain-independence.

**Strengths:**

This is an interesting problem, and could serve as the foundation for important neurosymbolic applications in the future.

The authors introduce OLLM, a general and scalable method that builds ontologies from scratch by fine-tuning an LLM. Instead of focusing on individual relations between entities, OLLM models entire subcomponents of the target ontology, reducing overfitting on high-frequency concepts through a custom regularizer. The principles of the approach are moderately innovative.

The paper also introduces new metrics for evaluating the quality of the generated ontology, measuring its semantic and structural similarity to the ground truth. This is laudable, but in going through the metrics, I noted some significant potential flaws that I note below.

**Weaknesses:**

I would question the validity of the  fuzzy F1 metric. Using embedding similarity is not a good fit here, as for difficult samples, it would likely be wrong. There is also bias that comes from pretraining (so the metric can't really be used in 'unusual' domains for which such embeddings are not available; but that's where the true applications of this proposal would be!) The gap between the literal and fuzzy F1 in the performance table should have been a warning to the authors.

The experiments could have been much stronger. A more difficult benchmark than either of the ones the authors selected would have been more valuable for assessing the true potential or limitations of the method.

The baselines could similarly have been stronger. It doesn't look like the authors considered some form of sophisticated (e.g., chain of thought) prompting, which could have made a big difference here.

**Questions:**

Given that there are so many ontologies and corpora out there, why only limit to these two very well-known datasets? I feel that the approach could have been better evaluated with multiple ontologies, some of which are unusual and hence not captured as well by LLMs.

**Limitations:**

The study focuses on constructing simple ontologies that include only concepts and taxonomic relations. To extend OLLM to produce non-taxonomic relations, tags indicating the relation type could be added to each edge during the linearization of subgraphs for sequence modeling. New evaluation metrics may also be needed to handle multiple types of relations. Another limitation is that the taxonomic relations in the generated ontologies are not always transitive due to the presence of cycles, a common issue in many ontology learning methods. There are existing algorithms for cycle removal to clean hierarchies. Additionally, the study could not fully control for data contamination since the pretraining dataset of Mistral 7B is not publicly known. However, the generated ontologies were found to be sufficiently different from the ground truth, suggesting that OLLM does not directly remember samples from its pretraining stage.

---

> ### Author Rebuttal · Authors · 2024-08-06
>
> We thank the reviewer for their detailed feedback.
>
> > Using embedding similarity is not a good fit here, as for difficult samples, it would likely be wrong. There is also bias that comes from pretraining (so the metric can't really be used in 'unusual' domains for which such embeddings are not available; but that's where the true applications of this proposal would be!)
>
> For the tasks considered in this paper, we did not observe instances where concepts with clearly different meanings were given similar embeddings. This can be credited to the quality of the pretrained embedding model and relatively common concepts present in Wikipedia and arXiv. Note that the pretraining of the embeddings model aims to cover every domain and to find the best generalising embeddings. We expect such embedding to remain informative even when applied out-of-distribution.
>
> We agree that for more exotic domains (e.g., protein ontology), a generic embedding model like the one used in this paper is unlikely to give accurate results. However, we argue that the evaluation framework proposed in this paper is still applicable as long as a more specialised embedder is available. We believe applications with accurate embeddings are much more prevalent than those with accurate ontologies.
>
> > The gap between the literal and fuzzy F1 in the performance table should have been a warning to the authors.
>
> We take this as an argument _for_ using embeddings. The cases where Literal and Fuzzy F1 disagree the most (e.g., when Literal F1 is comparatively high but Fuzzy F1 is comparatively low) are on methods that show signs of overfitting, particularly Memorisation and Finetuning. We observe Literal F1 has a strong bias towards methods that memorise the training set. This is because literal F1 is sensitive to semantically insignificant syntax differences such as casing and word form, whereas Fuzzy F1 (as well as Continuous F1 and Graph F1) are generally agnostic to syntax.
>
> We also emphasise that we do not claim that any single metric can truthfully reflect the quality of an ontology. Given the many aspects of what constitutes a good ontology, it is an oversimplification to summarise the performance with a single value. We thus use multiple metrics to get a holistic understanding of the output. From this perspective, it is not surprising that different metrics will arrive at different conclusions, otherwise it would not have been useful to use multiple metrics in the first place.
>
> > The experiments could have been much stronger. A more difficult benchmark than either of the ones the authors selected would have been more valuable for assessing the true potential or limitations of the method.
>
> We chose Wikipedia as our main benchmark as it covers a wide range of topics, so many specific domains of interest can likely be found as a subgraph of the Wikipedia ontology. We believe the diversity of topics makes it more challenging than specialised ontologies like WordNet (focusing on word-level relations) or GeoNames (focusing on Geography). We would like to know whether the reviewer has any particular benchmark in mind that they believe would be a strong addition to the paper. We are open to including more benchmarks if necessary.
>
> > The baselines could similarly have been stronger. It doesn't look like the authors considered some form of sophisticated (e.g., chain of thought) prompting, which could have made a big difference here.
>
> We designed our prompting baselines to study the merits and failure modes of using LLMs to build ontologies out-of-the-box. The main weakness of zero-shot prompting (as discussed in the paper) is the inability to produce ontologies that are structurally similar to the target. Many prompting techniques, such as chain-of-thought, focus on improving reasoning and logic (e.g., the original CoT paper) which do not appear to be the main bottleneck in ontology learning.
>
> We nonetheless thank the reviewer for this suggestion and have included an additional ablation baseline using a more deliberate prompting method (inspired by zero-shot CoT [1]). A detailed description of the experiment setup can be found in the general response. The results support our hypothesis above, showing no significant improvement over basic zero-shot prompting.
>
> > Given that there are so many ontologies and corpora out there, why only limit to these two very well-known datasets? I feel that the approach could have been better evaluated with multiple ontologies, some of which are unusual and hence not captured as well by LLMs.
>
> We designed OLLM to be very general and demonstrated its effectiveness on Wikipedia as a proof-of-concept, and on a different dataset (arXiv) to show its generalisation. We believe they are sufficient in answering our primary research question “How can LLMs be used effectively and scalably for OL?”. While we agree that it would be useful to study which domains are well-captured by LLMs, it is beyond the scope of this paper.
>
> However, as mentioned above, we are open to suggestions from the reviewer if they think a particular dataset would be a strong addition to the paper.
>
> ---
>
> Would the reviewer please consider raising the score if we have addressed their concerns?

---

> > ### Author Response · Authors · 2024-08-12
> >
> > Again, we thank the reviewer for their constructive feedback. As we are near the end of the discussion period, we would like to confirm if our answers are comprehensive and satisfactory. Should any new questions arise, we would be happy to answer them.

---

### Author Rebuttal · Authors · 2024-08-06

We thank all the reviewers for their time. The feedback is constructive and helpful.

We are happy to see that most reviewers found the end-to-end OL task interesting, and agree that our core contribution, OLLM, is a novel approach to leveraging LLMs for OL. Reviewers also agree that the paper is well-written.

Most of the reviewers’ concerns revolve around our experiment results and evaluation strategy, such as inconsistent ranking among different metrics, and insufficient baselines. We thank the reviewers for raising such points and in response have performed additional ablations and analysis. We give a summary of our new results here. Further discussions can be found in our response to each reviewer in the context of their concerns.

# Consistency

Reviewer 3YVJ questioned the consistency of the output ontology. For example, it does not guarantee an anti-symmetric relation under transitivity (i.e., there may be cycles). We analysed the ontology generated by OLLM and found that they are almost consistent, with only 97 simple cycles in Wikipedia and no cycles in arXiv. All the cycles in Wikipedia can be broken if we remove just 26 of the 10414 edges. This is quite surprising considering we did not explicitly optimise our model to satisfy consistency.

# Ablations

See supplementary pdf for full metrics.

## Chain of thought (CoT)

**Motivation**: Reviewer Qo2b suggested that a stronger baseline can be established if we employ more sophisticated prompting techniques, such as CoT.

**Method**: Prediction involves two rounds of inference: In the first round, we ask the model (Mistral 7B instruct) to describe the possible relevant concepts for the input document and to explain its reasoning. Then, we ask the model to predict the subgraph in the specified format given the additional, self-generated context.

**Result**: We tested the CoT method on Wikipedia and found no significant difference from basic zero-shot prompting.

**Interpretation**: Most advanced prompting techniques, including CoT, primarily aim to improve logic and reasoning. We hypothesise that the performance in OL is more dependent on the model’s understanding of natural language than its ability to perform multi-step reasoning, hence we do not observe any significant improvement from CoT.
## LLMs4OL

**Motivation**: Reviewer cxnM and 3YVJ both suggested that a LLM-based subtask composition baseline (as studied in LLMs4OL [2]) would be useful for evaluating whether the improvement by OLLM is due to the improved methodology (end-to-end modelling) or simply just because we used LLMs.

**Method**: The subtasks studied in LLMs4OL are concept discovery (given a document, predict its relevant concepts), and link prediction (given two concepts, predict whether they are taxonomically related). Unfortunately, constructing a baseline from such two subtasks is non-trivial. We encountered significant scalability issues in the link prediction stage as it required $O(n^2)$ inferences. We make two modifications to overcome such limitation:
1. After the concept discovery stage, we only discard all but the N most frequent concepts to limit the number of inferences required during link prediction, where N is the number of concepts in the ground truth.
2. Instead of using zero-shot Mistral 7B as the link predictor, we use a finetuned BERT as the link predictor as it runs much faster. Given that [2] demonstrated that finetuned models perform much better than zero-shot inference on link prediction, we expect the finetuned BERT to be at least as good, if not better, than zero-shot Mistral 7B on this subtask.

In summary, the method is as follows:
 1. Use zero-shot Mistral 7B for concept discovery, allowing it to tag more than one concept per document.
 2. Discard all but the top N most common concepts. Manually add the root concept.
 3. Perform link prediction with a finetuned BERT for all concept pairs.
 4. Apply post-processing, as described in section 3.2.

We design this baseline such that it is comparable to zero-shot end-to-end modelling: both use zero-shot Mistral 7B as the backbone, just utilised in different ways.

**Result**: We tested this method on Wikipedia and found that it is worse than zero-shot end-to-end modelling on all metrics except Motif Distance.

**Interpretation**: We take this as evidence that our end-to-end modelling approach is a clear improvement over traditional subtask-based OL. Not only does the link prediction-based method suffer from significant scalability bottlenecks for large ontologies, its performance is also worse. The results suggest that we can more effectively and efficiently leverage the capabilities of LLMs beyond just solving subtasks, such as by predicting subgraphs.

# Final remark
We ask the reviewers to evaluate our contribution in the context of existing methods for using LLMs in OL. Our experiments and ablations (above) suggest that our method is more effective and scalable than traditional LLM-based subtask composition methods (e.g., LLMs4OL). Additionally, in contrast to prior attempts to use LLMs in a more end-to-end manner that relies on qualitative evaluation [3, 4], our evaluation framework is more systematic and quantitative, laying the foundation for more rigorous research in the future. We hope the reviewers will consider raising the score if we have addressed any of their concerns.

[1]: Kojima, Takeshi, et al. "Large language models are zero-shot reasoners."

[2]: Babaei Giglou, Hamed, Jennifer D’Souza, and Sören Auer. "LLMs4OL: Large language models for ontology learning."

[3]: Funk, Maurice, et al. "Towards ontology construction with language models."

[4]: Trajanoska, Milena, Riste Stojanov, and Dimitar Trajanov. "Enhancing knowledge graph construction using large language models."

---

### Decision · Program_Chairs · 2024-09-25

**Decision:**

Accept (poster)

**Comment:**

The paper provides a novel approach for ontology learning using LLMs with a novel regularizer and a novel suite of evaluation metrics. Those were acknowledged as strengths by the reviewers. Concerns in the reviews are the limited expressivity of the learned ontology and the breadth of the evaluation. Some clarifications demanded by the reviewers have been added in the rebuttal stage increasing my confidence in the paper. Overall, the paper presents a novel and interesting approach advancing the state-of-the-art in ontology learning. Therefore, I recommend acceptance.